# HyperDomainNet: Universal Domain Adaptation for Generative Adversarial Networks

**Aibek Alanov**[*][†]
HSE University, AIRI[‡]
Moscow, Russia
alanov.aibek@gmail.com

**Vadim Titov**[*][†]
MIPT,[§] AIRI[‡]
Moscow, Russia
titov.vn@phystech.edu

**Dmitry Vetrov**
HSE University, AIRI[‡]
Moscow, Russia
vetrovd@yandex.ru

## Abstract

Domain adaptation framework of GANs has achieved great progress in recent years as a main successful approach of training contemporary GANs in the case of very limited training data. In this work, we significantly improve this framework by proposing an extremely compact parameter space for fine-tuning the generator. We introduce a novel domain-modulation technique that allows to optimize only 6 thousand-dimensional vector instead of 30 million weights of StyleGAN2 to adapt to a target domain. We apply this parameterization to the state-of-art domain adaptation methods and show that it has almost the same expressiveness as the full parameter space. Additionally, we propose a new regularization loss that considerably enhances the diversity of the fine-tuned generator. Inspired by the reduction in the size of the optimizing parameter space we consider the problem of multi-domain adaptation of GANs, i.e. setting when the same model can adapt to several domains depending on the input query. We propose the HyperDomainNet that is a hypernetwork that predicts our parameterization given the target domain. We empirically confirm that it can successfully learn a number of domains at once and may even generalize to unseen domains. Source code can be found at this github repository.

## 1 Introduction

Contemporary generative adversarial networks (GANs) [8, 14, 15, 13, 3] show remarkable performance in modeling image distributions and have applications in a wide range of computer vision tasks (image enhancement [18, 42], editing [9, 31], image-to-image translation [12, 46, 47], etc.). However, the training of modern GANs requires thousands of samples that limits its applicability only to domains that are represented by a large set of images. The mainstream approach to sidestep this limitation is transfer learning (TL), i.e. fine-tuning the generative model to a domain with few samples starting with a pretrained source model.

The standard approach of GAN TL methods is to fine-tune almost *all* weights of the pretrained model [19, 22, 38, 37, 13, 44, 24, 6, 48]. It can be reasonable in the case when the target domain is very far from the source one, e.g. when we adapt the generator pretrained on human faces to the domain of animals or buildings. However, there is a wide range of cases when the distance between data domains is not so far. In particular, the majority of target domains used in works [19, 37, 24, 6, 48] are similar to the source one and differ mainly in texture, style, geometry while keep the same content like faces or outdoor scenes. For such cases it seems redundant to fine-tune all weights of the source

---

[*]Equal contribution
[†]Corresponding author
[‡]Artificial Intelligence Research Institute
[§]Moscow Institute of Physics and Technology

36th Conference on Neural Information Processing Systems (NeurIPS 2022).

generator. It was shown in the paper [40] that after transfer learning of the StyleGAN2 [15] to similar domains some parts of the network almost do not change. This observation motivates us to find a more efficient and compact parameter space for domain adaptation of GANs.

In this paper, we propose a novel *domain-modulation* operation that reduces the parameter space for fine-tuning the StyleGAN2. The idea is to optimize for each target domain only a single vector $d$. We incorporate this vector into the StyleGAN2 architecture through the modulation operation at each convolution layer. The dimension of the vector $d$ equals 6 thousand that is *5 thousand times* less than the original weights space of the StyleGAN2. We apply this parameterization for the state-of-the-art domain adaptation methods StyleGAN-NADA [6] and MindTheGAP [48]. We show that it has almost the same expressiveness as the full parameterization while being more lightweight. To further advance the domain adaptation framework of GANs we propose a new regularization loss that improves the diversity of the fine-tuned generator.

Such considerable reduction in the size of the proposed parameterization motivates us to consider the problem of multi-domain adaptation of GANs, i.e. when the same model can adapt to multiple domains depending on the input query. Typically, this problem is tackled by previous methods just by fine-tuning separate generators for each target domain independently. In contrast, we propose to train a hyper-network that predicts the vector $d$ for the StyleGAN2 depending on the target domain. We call this network as HyperDomainNet. Such hyper-network would be impossible to train if we needed to predict all weights of StyleGAN2. The immediate benefits of multi-domain framework consist of reducing the training time and the number of trainable parameters because instead of fine-tuning $n$ separate generators we train one HyperDomainNet to adapt to $n$ domains simultaneously. Another advantage of this method is that it can generalize to *unseen* domains if $n$ is sufficiently large and we empirically observe this effect.

We provide extensive experiments to empirically confirm the effectiveness of the proposed parameterization and the regularization loss on a wide range of domains. We illustrate that our parameterization can achieve quality comparable with the full parameterization (i.e. when we optimize all weights). The proposed regularization loss significantly improves the diversity of the fine-tuned generator that is validated qualitatively and quantitatively. Further, we conduct experiments with the HyperDomainNet and show that it can be successfully trained on a number of target domains simultaneously. Also we show that it can generalize to a number of diverse unseen domains.

To sum up, our main contributions are

- We reduce the number of trainable parameters for domain adaptation of StyleGAN2 [15] generator by proposing the domain-modulation technique. Instead of fine-tuning all 30 millions weights of StyleGAN2 for each new domain now we can train only 6 thousand-dimensional vector.

- We introduce a novel regularization loss that considerably improves the diversity of the adapted generator.

- We propose a HyperDomainNet that predicts the parameterization vector for the input domain and allows multi-domain adaptation of GANs. It shows inspiring generalization results on unseen domains.

## 2 Related Work

**Domain Adaptation of GANs**   The aim of few-shot domain adaptation of GANs is to learn accurate and diverse distribution of the data represented by only a few images. The standard approach is to utilize a generator pretrained on source domain and fine-tune it to a target domain. There are generally two different regimes of this task. The first one is when we adapt the generator to completely new data (e.g. faces → landscapes, churches, etc.), and the second regime is when the target domain relates to the source one (e.g. faces → sketches, artistic portraits, etc.).

Methods that tackle the first regime typically require several hundreds or thousands samples to adapt successfully. Such setting assumes that the weights of the generator should be changed significantly because the target domain can be very far from the source. The paper [13] shows that for distant domains training from scratch gives comparable results to transfer learning. It also confirms that for such regime there is no point to reduce the parameter space. Typcially such approaches utilize data augmentations [13, 33, 44, 45], or use auxiliary tasks for the discriminator to more accurately fit the

available data [20, 41], or freeze lower layers of the discriminator to avoid overfitting [22]. Another standard techniques for the effective training of GANs is to apply different normalization methods [21, 16, 2] to stabilize the training process.

In the second regime the transfer learning is especially crucial because the pretrained generator already contains many information about the target domain. In this setting the required number of available data can be significantly smaller and range from hundreds to several images. The main challenges in the case of such limited data are to avoid over-fitting of the generator and leverage its diversity learned from the source domain. To tackle these challenges existing methods introduce restrictions on the parameter space [29, 23], mix the weights of the adapted and the source generators [26], utilize a small network to force sampling in special regions of the latent space [37], propose new regularization terms [19, 34], or apply contrastive learning techniques to enhance cross-domain consistency [24]. The state-of-the-art methods [6, 48] leverage supervision from vision-language CLIP model [27]. StyleGAN-NADA [6] applies it for text-based domain adaptation when we have no access to images but only to the textual description. MindTheGap [48] employs CLIP model to further significantly improve the quality of one-shot domain adaptation.

**Constraining Parameter Space for GAN's Adaptation**    In the second regime of GAN's adaptation it is especially important for the generator to leverage the information from the source domain during adapting to the target one. The common approach is to introduce some restrictions on the trainable weights to regularize them during fine-tuning. For example, the work [29] proposes to optimize only the singular values of the pretrained weights and apply it for few shot domain adaptation, however the reported results show the limited expressiveness of such parameterization [29]. Another method [23] constrains the parameter space for models with batch normalization (BN) layers such as BigGAN [3] by optimizing only BN statistics during fine-tuning. While it allows to decrease the number of trainable parameters, it also considerably reduces the expressiveness of the generator [29, 24]. Other approach is to adaptively choose a subset of layers during optimization at each step as in StyleGAN-NADA [6]. It helps to stabilize the training, however it does not reduce the parameter space because each layer can potentially be fine-tuned. In contrast, the size of our parameterization is less by orders of magnitude than the size of the full parameter space while having the comparable expressiveness.

## 3   Preliminaries

In this work, we focus on StyleGAN generators in the context of domain adaptation. We consider StyleGAN2 [15] as a base model. As the state-of-the-art domain adaptation methods we use StyleGAN-NADA [6] and MindTheGAP [48].

**StyleGAN2**    The StyleGAN2 generation process consists of several components. The first part is a mapping network $M(z)$ that takes as an input random vectors $z \in \mathcal{Z}$ from the initial latent space, $\mathcal{Z}$ that is typically normally distributed. It transforms these vectors $z$ into the intermediate latent space $\mathcal{W}$. Each vector $w \in \mathcal{W}$ is further fed into different affine transformations $A(w)$ for each layer of the generator. The output of this part forms StyleSpace $\mathcal{S}$ [39] that consists of channel-wise style parameters $s = A(w)$. The next part of the generation process is the synthesis network $G_{sys}$ that takes as an input the constant tensor $c$ and style parameters $s$ at the corresponding layers and produces the final feature maps at different resolutions $F = G_{sys}(c, s)$. These feature maps move on to the last part which consists of toRGB layers $G_{tRGB}$ that generate the output image $I = G_{tRGB}(F)$.

**Problem Formulation of Domain Adaptation**    The problem of domain adaptation of StyleGAN2 can be formulated as follows. We are given a trained generator $G^A$ for the source domain $A$, and the target domain $B$ that is represented by the one image $I_B$ (one-shot adaptation) or by the text description $t_B$ (text-guided adaptation). The aim is to fine-tune the weights $\theta$ of a new generator $G_\theta^B$ for the domain $B$ starting from the weights of $G^A$. The optimization process in the general form is

$$\mathcal{L}_B(\theta) = \mathcal{L}(\{G_\theta^B(w_i)\}_{i=1}^n, \{G^A(w_i)\}_{i=1}^n, G_\theta^B, B, A) \rightarrow \min_\theta, \qquad (1)$$

where $\mathcal{L}$ is some loss function, $n$ is a batch size, $w_1, \ldots, w_n$ are random latent codes, $\{G_\theta^B(w_i)\}_{i=1}^n$ and $\{G^A(w_i)\}_{i=1}^n$ are batches of images sampled by $G_\theta^B$ and $G^A$ generators, respectively, and $B, A$ are domains that are represented by images or text descriptions.

**CLIP model** CLIP [27] is a vision-language model that is composed of text and image encoders $E_T$, $E_I$, respectively, that maps their inputs into a joint, multi-modal space of vectors with a unit norm (this space is often called as CLIP space). In this space the cosine distance between embeddings reflects the semantic similarity of the corresponding objects.

**StyleGAN-NADA** StyleGAN-NADA [6] is a pioneering work that utilizes the CLIP model [27] for text-guided domain adaptation of StyleGAN. The proposed loss function is

$$\Delta T(B, A) = E_T(t_B) - E_T(t_A),$$
$$\Delta I(G_\theta^B(w), G^A(w)) = E_I(G_\theta^B(w)) - E_I(G^A(w)),$$
$$\mathcal{L}_{direction}(G_\theta^B(w), G^A(w), B, A) = 1 - \frac{\Delta I(G_\theta^B(w), G^A(w)) \cdot \Delta T(B, A)}{|\Delta I(G_\theta^B(w), G^A(w))||\Delta T(B, A)|}. \tag{2}$$

The idea is to align the CLIP-space direction between the source and target images $\Delta I(G_\theta^B(w), G^A(w))$ with the direction between a pair of source and target text descriptions $\Delta T(B, A)$. So, the overall optimization process has the form

$$\mathcal{L}_B(\theta) = \sum_{i=1}^n \mathcal{L}_{direction}(G_\theta^B(w_i), G^A(w_i), B, A) \rightarrow \min_\theta. \tag{3}$$

In StyleGAN-NADA method the $\mathcal{L}_B(\theta)$ loss is optimized only with respect to the weights $\theta$ of the synthesis network $G_{sys}^B$ which has 24 million weights.

**MindTheGap** The MindTheGap method [48] is proposed for a one-shot domain adaptation of StyleGAN, i.e. the domain $B$ is represented by the single image $I_B$. In principle StyleGAN-NADA method can solve this problem just by replacing the text direction $\Delta T(B, A)$ from Equation (2) to an image one

$$\Delta I'(B, A) = E_I(I_B) - \frac{1}{|A|} \sum_{I_A \in A} [E_I(I_A)], \tag{4}$$

where $\frac{1}{|A|} \sum_{I_A \in A} [E_I(I_A)]$ is the mean embedding of the images from domain $A$. However, as stated in [48] this leads to an undesirable effect that transferred images lose the initial diversity of domain $A$ and become too close to the $I_B$ image. So, the key idea of the MindTheGap is to replace the mean embedding from Equation (4) by the embedding of projection $I_A^*$ of $I_B$ image to $A$ domain obtained by the GAN inversion method II2S [49]:

$$\Delta I''(B, A) = E_I(I_B) - E_I(I_A^*), \tag{5}$$

So, the MindTheGap uses the modified $\mathcal{L}'_{direction}$ loss that is renamed to $\mathcal{L}_{clip\_accross}$

$$\mathcal{L}_{clip\_accross}(G_\theta^B(w), G^A(w), B, A) = 1 - \frac{\Delta I(G_\theta^B(w), G^A(w)) \cdot \Delta I''(B, A)}{|\Delta I(G_\theta^B(w), G^A(w))||\Delta I''(B, A)|}. \tag{6}$$

In addition to this idea several new regularizers are introduced that force the generator $G_\theta^B$ to reconstruct the $I_B$ image from its projection $I_A^*$. It further stabilizes and improves the quality of domain adaption. Overall, the MindTheGAP loss function $\mathcal{L}_{MTG}$ has four terms to optimize $G_\theta^B$. For more details about each loss please refer to the original paper [48].

## 4 Approach

### 4.1 Domain-Modulation Technique for Domain Adaptation

Our primary goal is to improve the domain adaptation of StyleGAN by exploring an effective and compact parameter space to use it for fine-tuning $G_\theta^B$. As we described in Section 3 StyleGAN has four components: the mapping network $M(\cdot)$, affine transformations $A(\cdot)$, the synthesis network $G_{sys}(\cdot, \cdot)$, and toRGB layers $G_{tRGB}(\cdot)$. It is observed in the paper [40] that the main part of StyleGAN that is mostly changed during fine-tuning to a target domain is the synthesis network

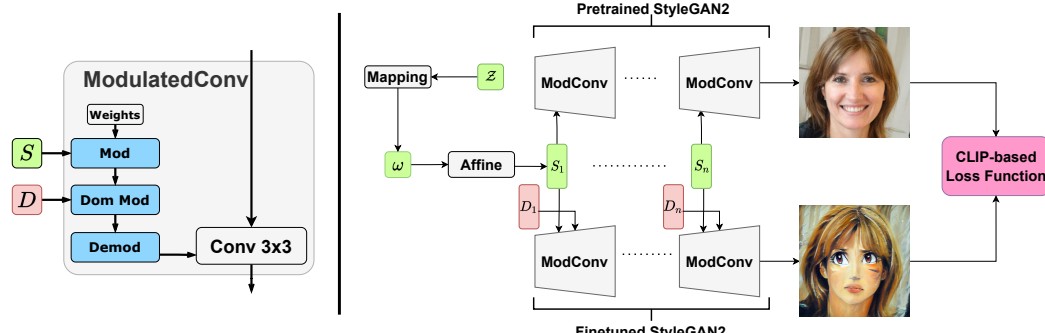

(a) Domain-modulation technique    (b) Fine-tuning StyleGAN2 by optimizing the domain vector D

Figure 1: Detailed diagram of proposed method. (a) Revised ModulatedConv block with introduced domain-modulation operation. (b) Fully detailed training process of the domain adaptation with the proposed domain-modulation technique.

$G_{sys}(\cdot, \cdot)$. It is also confirmed by StyleGAN-NADA [6] and MindTheGap [48] methods as they adapt only the weights of $G_{sys}(\cdot, \cdot)$ for the target domain.

So, we aim to find an effective way to fine-tune the weights of feature convolutions of $G_{sys}(\cdot, \cdot)$. In StyleGAN2 [15] these convolutions utilize modulation/demodulation operations to process the input tensor and the corresponding style parameters $s$. Let us revisit the mechanism of these operations:

$$\text{modulation: } w'_{ijk} = s_i \cdot w_{ijk}, \tag{7}$$

$$\text{demodulation: } w''_{ijk} = \frac{w'_{ijk}}{\sqrt{\sum_{i,k} {w'_{ijk}}^2 + \varepsilon}}, \tag{8}$$

where $w, w'$ and $w''$ are the original, modulated and demodulated weights, respectively, $s_i$ is the component of the style parameters $s$, $i$ and $j$ enumerate input and output channels, respectively. The idea behind modulation/demodulation is to replace the standard adaptive instance normalization (AdaIN) [35, 5] to a normalization that is based on the expected statistics of the input feature maps rather than forcing them explicitly [15]. So, the modulation part is basically an adaptive scaling operation as in AdaIN that is controlled by the style parameters $s$. This observation inspires us to use this technique for the domain adaptation.

The problem of fine-tuning GANs to a new domain is very related to the task of style transfer where the goal is also to translate images from the source domain to a new domain with the specified style. The contemporary approach to solve this task is to train an image-to-image network which takes the target style as an input condition. The essential ingredient of such methods is the AdaIN that provides an efficient conditioning mechanism. In particular, it allows to train arbitrary style transfer models [11]. So, it motivates us to apply the AdaIN technique for adapting GANs to new domains.

We introduce a new *domain-modulation* operation that reduces the parameter space for fine-tuning StyleGAN2. The idea is to optimize only a vector $d$ with the same dimension as the style parameters $s$. We incorporate this vector into StyleGAN architecture by the additional modulation operation after the standard one from Equation (7):

$$\text{domain-modulation: } w'_{ijk} = d_i \cdot w_{ijk}, \tag{9}$$

where $d_i$ is the component of the introduced domain parameters $d$ (see Figure 1a). So, instead of optimizing all weights $\theta$ of the $G_{sys}$ part we train only the vector $d$.

We apply these new parameterization to StyleGAN-NADA and MindTheGAP methods, i.e. instead of optimizing its loss functions wrt $\theta$ we optimize it wrt $d$ vector (see Figure 1b) The dimension of the vector $d$ equals 6 thousand that is 4 thousand times less than the original weights space $\theta$ of $G_{sys}(\cdot, \cdot)$ part. While the proposed parameter space is radically more constrained we observe that it has the expressiveness comparable with the whole weight space.

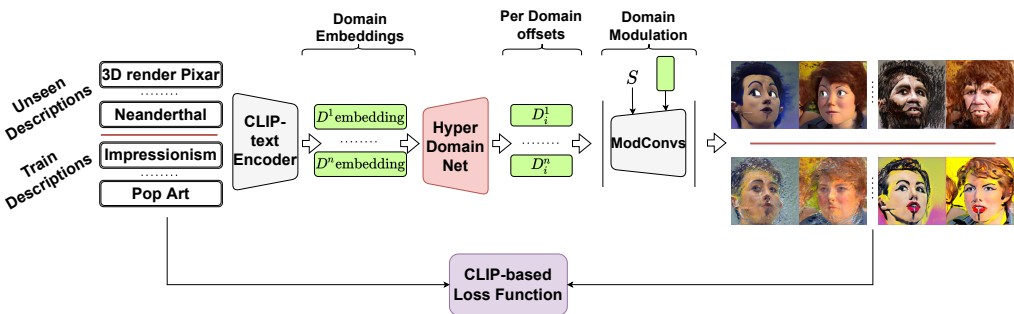

Figure 2: Detailed training process of the HyperDomainNet. On the training phase only reference descriptions are included into CLIP-guided training.

## 4.2 Improving Diversity of CLIP-Guided Domain Adaptation

The CLIP-based domain adaptation methods StyleGAN-NADA and MindTheGap use $\mathcal{L}_{direction}$ (or $\mathcal{L}_{clip\_accross}$) loss (see Equations (2) and (6)) that was initially introduced to deal with the mode collapsing problem of the fine-tuned generator [6]. However, we empirically observe that it solves the issue only partially. In particular, it preserves the diversity only at the beginning of the fine-tuning process and starts collapsing after several hundred iterations. It is a significant problem because for some domains we need much more iterations to obtain the acceptable quality.

The main cause of such undesirable behaviour of the $\mathcal{L}_{direction}$ (the same for $\mathcal{L}_{clip\_accross}$) loss is that it calculates the CLIP cosine distance between embeddings that do not lie in the CLIP space. Indeed, the cosine distance is a natural distance for objects that lie on a CLIP sphere but becomes less evident for vectors $\Delta T, \Delta I$ that represent the difference between clip embeddings that no longer lie on a unit sphere. Therefore, the idea behind the $\mathcal{L}_{direction}$ loss may be misleading and in practice we can observe that it still suffers from mode collapse.

We introduce a new regularizer for improving diversity that calculates the CLIP cosine distance only between clip embeddings. We called it *indomain angle consistency* loss and we define it as follows

$$\mathcal{L}_{indomain-angle}(\{G_d^B(w_i)\}_{i=1}^n, \{G^A(w_i)\}_{i=1}^n, B, A) = \tag{10}$$

$$= \sum_{i,j}^n (\langle E_I(G^A(w_i)), E_I(G^A(w_j)) \rangle - \langle E_I(G_d^B(w_i)), E_I(G_d^B(w_j)) \rangle)^2, \tag{11}$$

The idea of $\mathcal{L}_{indomain-angle}$ loss is to preserve the CLIP pairwise cosine distances between images before and after domain adaptation. We observe that this loss significantly improves the diversity of the generator $G_d^B$ compared to the original $L_{direction}$ or $\mathcal{L}_{clip\_accross}$ losses.

## 4.3 Designing the HyperDomainNet for Universal Domain Adaptation

The proposed domain-modulation technique allows us to reduce the number of trainable parameters which motivates us to tackle the problem of multi-domain adaption of StyleGAN2. Our aim is to train the HyperDomainNet that predicts the domain parameters given the input target domain. This problem can be formulated as follows. We are given a trained generator $G^A$ for a source domain $A$ and a number of target domains $B_1, \ldots, B_m$ that can be represented by the single image or the text description. The aim is to learn the HyperDomainNet $D_\varphi(\cdot)$ that can predict the domain parameters $d_{B_i} = D_\varphi(B_i)$ which will be used to obtain the fine-tuned generator $G_{d_{B_i}}^{B_i}$ by the domain-modulation operation (see Section 4.1).

In this work, we focus on the setting when the target domains $B_1, \ldots, B_m$ are represented by text descriptions $t_{B_1}, \ldots, t_{B_m}$. The HyperDomainNet $D_\varphi(\cdot)$ takes as an input the embedding of the text obtained by the CLIP encoder $E_T(\cdot)$ and outputs the domain parameters $d_{B_i} = D_\varphi(E_T(t_{B_i}))$. The training process is described in the Figure 2.

To train the HyperDomainNet $D_\varphi(\cdot)$ we use the sum of $\mathcal{L}_{direction}$ losses for each target domains. In addition, we introduce $\mathcal{L}_{tt-direction}$ loss ("tt" stands for target-target) that is the same as $\mathcal{L}_{direction}$,

StyleGAN-NADA                                    Ours

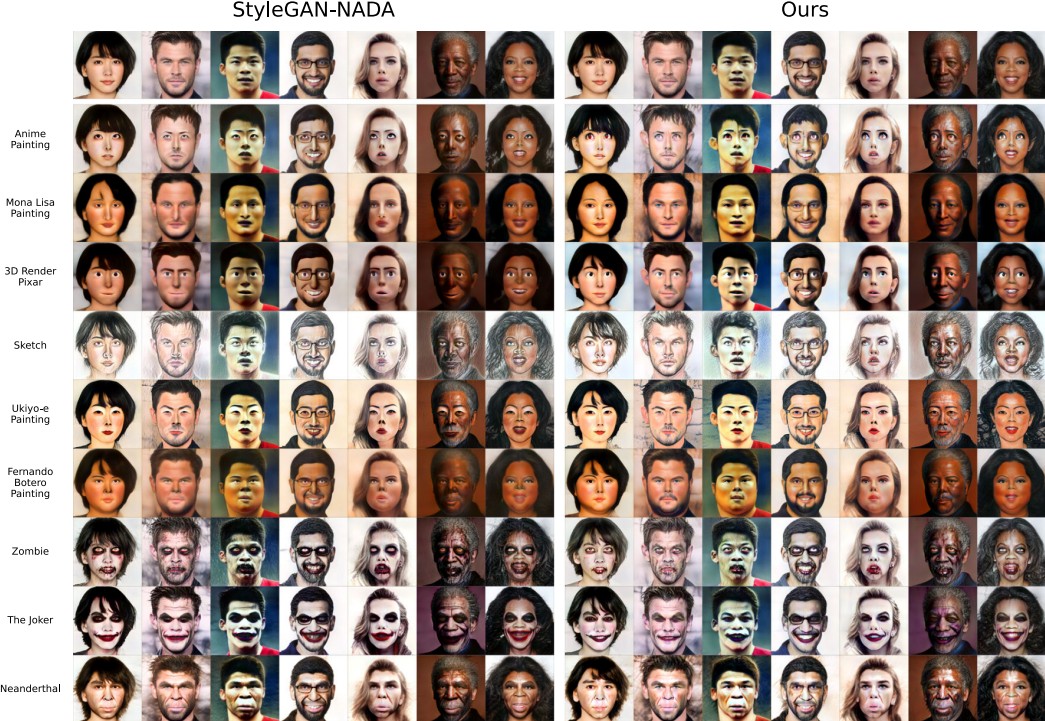

Figure 3: Comparison with the original StyleGAN-NADA [6] method (left) and its version with our parameterization.

but we compute it between two target domains instead of target and source. The idea is to keep away the images from different target domains in the CLIP space. We observe that without $\mathcal{L}_{tt-direction}$ loss the HyperDomainNet tends to learn the mixture of domains.

In multi-domain adaptation setting, the regularizer $\mathcal{L}_{indomain-angle}$ becomes inefficient because during training batch consists of samples from different domains and the number of images from one domain can be very small. Therefore, we introduce an alternative regularization $\mathcal{L}_{domain-norm}$ for the HyperDomainNet that constrains the norm of the predicted domain parameters. To be exact it equals to $\|D_{\varphi}(E_T(t_{B_i})) - 1\|^2$.

So, the objective function of the HyperDomainNet consists of $\mathcal{L}_{direction}$, $\mathcal{L}_{tt-direction}$ and $\mathcal{L}_{domain-norm}$ losses. For more detailed description of these losses the overall optimization process, please refer to Appendix A.2.

## 5   Experiments

In this section, we provide qualitative and quantitative results of the proposed approaches. At first, we consider the text-based domain adaptation and show that our parameterization has comparable quality with the full one. Next, we tackle one-shot domain adaptation and confirm the same quantitatively and also show the importance of the $\mathcal{L}_{indomain-angle}$ loss. Finally, we solve the multi-domain adaptation problem by the proposed HyperDomainNet, show its generalization ability on unseen domains. For the detailed information about setup of the experiments please refer to Appendix A.1.

**Text-Based Domain Adaptation**   We compare the StyleGAN-NADA [6] method with the proposed parameterization and the original version on a number of diverse domains. In Figure 3, we see that the expressiveness of our parameterization is on par with the original StyleGAN-NADA. We observe that the domain-modulation technique allows to adapt the generator to various style and texture changes. For results on more domains please refer to Appendix A.3. We also provide quantitative results for

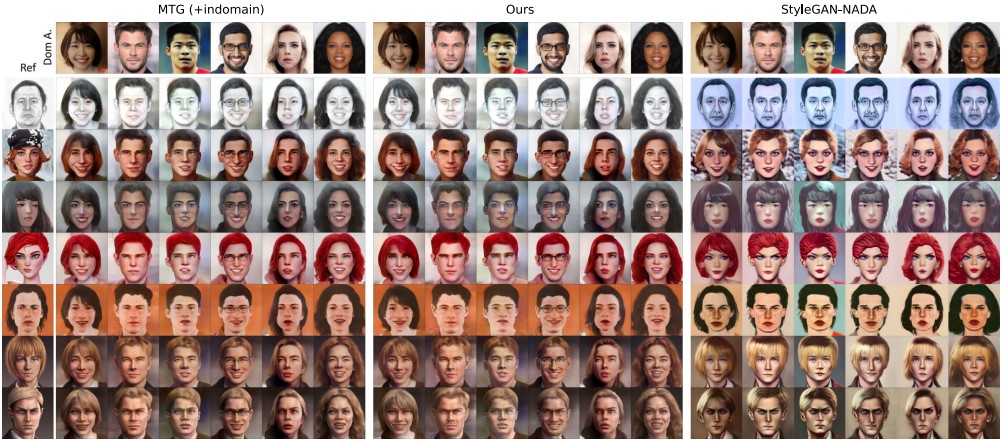

Figure 4: Comparison with one-shot domain adaptation methods. Left block is MindThe-Gap+indomain and right block is StyleGAN-NADA [48]. The middle block is the MindThe-Gap+indomain with our parameterization.

this setting in Appendix A.3.3 which show that our parameterization has the comparable performance as the full one.

**One-Shot Domain Adaptation**    In this part, we examine our parameterization and the indomain angle consistency loss by applying them to the MindTheGap [48] method. We show qualitative and quantitative results and compare them with other few-shot domain adaptation methods such as StyleGAN-NADA, TargetCLIP [4] and Cross-correspondence [24] method. To assess the domain adaptation quality we use the standard metrics FID [10], precision and recall [17]. As a target domain we take the common benchmark dataset of face sketches [36] that has approximately 300 samples. We consider the one-shot adaptation setting. We provide the results in Table 1. At fisrt, we see that the MindTheGap with our parameterization shows comparable results with the original version while having less trainable parameters by three orders of magnitude. Secondly, we examine the effectiveness of the indomain angle consistency. We show that it considerably improves FID and precision metrics for both the original MindTheGap and the one with our parameterization.

The qualitative results are provided in Figure 4 for MindTheGap+indomain, MindTheGap+indomain with our parameterization ("Ours") and StyleGAN-NADA. For other methods please see Appendix A.4. We observe that MindTheGap+indomain and our version shows comparable visual quality and outperform StyleGAN-NADA in terms of diversity and maintaining the similarity to the source image.

Overall, we demonstrate that our parameterization is applicable to the state-of-the-art methods StyleGAN-NADA and MindTheGap and it can be further improved by the indomain angle consistency loss.

**Multi-Domain Adaptation**    Now we consider the multi-domain adaptation problem. We apply the HyperDomainNet in two different scenarios: (i) training on fixed number of domains, (ii) training on potentially arbitrary number of domains. The first scenario is simple, we train the HyperDomainNet on 20 different domains such as "Anime Painting", "Pixar", etc. (for the full list of domains please refer to Appendix A.2.4). The second scheme is more complicated. We fix large number of domains (several hundreds) and calculate its CLIP embeddings. During training we sample new embeddings from the convex hull of the initial ones and use them in the optimization process (see Figure 2). This technique allows us to generalize to unseen domains. For more details about both scenarios please refer to Appendix A.2.

The results of the HyperDomainNet for both scenarios are provided in Figure 5. The left part is results for the first setting, the right one is results for the unseen domains in the second scheme. For more domains and generated images please refer to Appendix A.2. We see that in the first scenario the Hy-

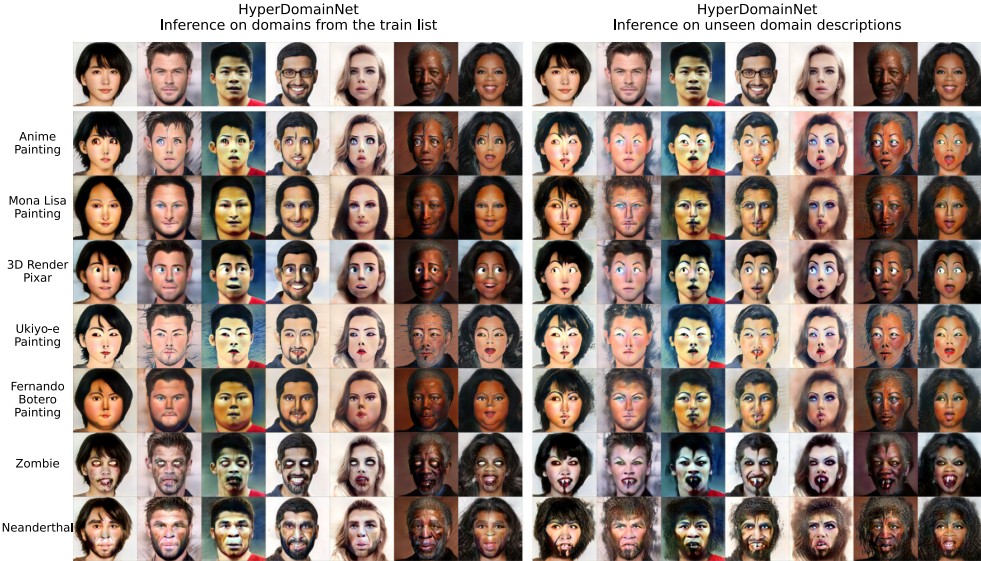

Figure 5: Comparison of training setups. The top row represents the real images embedded into StyleGAN2 latent space which latents are then used for HyperDomainNet inference. The left block represents results obtained from text-descriptions presented in the train list. The right block represents results of HyperDomainNet inference on unseen text-descriptions.

perDomainNet shows results comparable to the case when we train separate models for each domain (see Figure 3). It shows that the proposed optimization process for the HyperDomainNet is effective. The results for the second scenario looks promising. We can observe that the HyperDomainNet has learnt very diverse domains and shows sensible adaptation results for unseen ones.

We also provide an ablation study on the loss terms we use for training of the HyperDomainNet in Appendix A.2.6. It demonstrates quantitatively and qualitatively that the proposed losses are essential for the effective training of the HyperDomainNet in the setting of the multi-domain adaptation problem.

## 6    Conclusion

We propose a novel domain-modulation technique that allows us to considerably reduce the number of trainable parameters during domain adaptation of StyleGAN2. In particular, instead of fine-tuning almost all 30 million weights of the StyleGAN2 we optimize only 6 thousand-dimensional domain vector. We successfully apply this technique to the state-of-the-art text-based and image-based

Table 1: Evaluation of one-shot adaptation methods. Results for TargetCLIP, Cross-correspondence and StyleGAN-NADA methods are taken from [48].

| Model | Model quality | | | Model complexity |
|---|---|---|---|---|
| | FID | Precision | Recall | # trainable parameters |
| TargetCLIP [4] | 199.33 | 0.000 | 0.293 | 30M |
| Cross-correspondence [24] | 158.86 | 0.001 | 0 | 30M |
| StyleGAN-NADA [6] | 124.55 | 0.118 | 0 | 24M |
| MindTheGap [48] | 78.35 | 0.326 | 0.017 | 24M |
| MindTheGap (our param.) | 79.83 | 0.452 | 0.017 | 6k |
| MindTheGap+indomain | 71.46 | 0.503 | 0.014 | 24M |
| MindTheGap+indomain (our param.) | 72.71 | 0.472 | 0.028 | 6k |

domain adaptation methods. We show quantitatively and qualitatively that it can achieve the same quality as optimizing all weights of the StyleGAN2.

To deal with the mode collapsing problem of the domain adaptation methods we introduce a new indomain angle consistency loss $\mathcal{L}_{indomain-angle}$ that preserves the CLIP pairwise cosine distances between images before and after domain adaptation. We demonstrate that it improves the diversity of the fine-tuned generator both for text-based and one-shot domain adaptation.

We also consider the problem of multi-domain adaptation of StyleGAN2 when we aim to adapt to several domains simultaneously. Before our proposed parameterization it was infeasible because we should predict all weights of StyleGAN2 for each domain. Thanks to our efficient parameterization we propose HyperDomainNet that predicts the 6 thousand-dimensional domain vector $d$ for the Style-GAN2 given the input domain. We empirically show that it can be trained to 20 domains successfully which is the first time when the StyleGAN2 was adapted to several domains simultaneously. We also train the HyperDomainNet for the large number of domains (more than two hundred) with applying different augmentations to the domain descriptions (see details in Appendix A.2). We demonstrate in practice that in such setting the HyperDomainNet can generalize to unseen domains.

**Limitations and societal impact**   The main limitation of our approach is that it is not applicable for the cases when target domains are very far from the source one. In such setting, we cannot limit the parameter space, so we should use the full parameterization.

The potential negative societal impacts of domain adaptation of GANs and generally training of GANs include different forms of disinformation, e.g. deepfakes of celebrities or senior officials, fake avatars in social platforms. However, it is the issue of the whole field and this work does not amplify this impact.

# 7   Acknowledgments and Disclosure of Funding

The publication was supported by the grant for research centers in the field of AI provided by the Analytical Center for the Government of the Russian Federation (ACRF) in accordance with the agreement on the provision of subsidies (identifier of the agreement 000000D730321P5Q0002) and the agreement with HSE University No. 70-2021-00139. Additional revenues of the authors for the last three years: laboratory sponsorship by Samsung Research, Samsung Electronics and Huawei Technologies; Institute for Information Transmission Problems, Russian Academy of Science.

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
