# A    Appendix

## A.1    Setup of the Experiments

### A.1.1    Implementation Details

We implement our experiments using PyTorch[5] deep learning framework. For StyleGAN2 [15] architecture we use the popular PyTorch implementation [6]. We attach all source code that reproduces our experiments as a part of the supplementary material. We also provide configuration files to run each experiment.

### A.1.2    Datasets

We use source StyleGAN2 models pretrained on the following datasets: (i) Flickr-Faces-HQ (FFHQ) [14], (ii) LSUN Church, (iii) LSUN Cars, and (iv) LSUN Cats [43]. As target domains we mainly use the text descriptions from [6] and style images from [48]. For quantitative comparison with other methods we use face sketches [36] as the standard dataset for domain adaptation.

### A.1.3    Licenses and Data Privacy

Tables 2, 3 provide sources and licenses of the models and datasets we used in our work.

Table 2: Models used in our work, their sources and licenses.

| Model | Source | License |
|---|---|---|
| StyleGAN2 | [15] | Nvidia Source Code License-NC |
| pSp | [28] | MIT License |
| e4e | [32] | MIT License |
| ReStyle | [1] | MIT License |
| StyleCLIP | [25] | MIT License |
| CLIP | [27] | MIT License |
| StyleGAN2-pytorch | [30] | MIT License |
| StyleGAN-ADA | [13] | Nvidia Source Code License |
| Cross-correspondence | [24] | Adobe Research License |

Table 3: Datasets used in our work, their sources and licenses.

| Dataset | Source | License |
|---|---|---|
| FFHQ | [14] | CC BY-NC-SA 4.0[7] |
| LSUN | [43] | No License |
| Sketches | [24] | Adobe Research License |

### A.1.4    Total Amount of Compute Resources

We run our experiments on Tesla A100 GPUs. We used approximately 12000 GPU hours to obtain the reported results and for intermediate experiments.

---

[5]https://pytorch.org
[6]https://github.com/rosinality/stylegan2- pytorch

## A.2 Training of the HyperDomainNet (HDN)

### A.2.1 Training Losses

As we describe in Section 4.3 we train HDN $D_\varphi(\cdot)$ using three losses $\mathcal{L}_{direction}$, $\mathcal{L}_{tt-direction}$, and $\mathcal{L}_{domain-norm}$. Each loss is defined as follows:

$$\mathcal{L}_{direction}(G^{B_i}_{d_{B_i}}(w), G^A(w), B_i, A) = 1 - \frac{\Delta I(G^{B_i}_{d_{B_i}}(w), G^A(w)) \cdot \Delta T(B_i, A)}{|\Delta I(G^{B_i}_{d_{B_i}}(w), G^A(w))||\Delta T(B_i, A)|}, \tag{12}$$

$$\mathcal{L}_{tt-direction}(G^{B_i}_{d_{B_i}}(w), G^{B_j}_{d_{B_j}}(w), B_i, B_j) = 1 - \frac{\Delta I(G^{B_i}_{d_{B_i}}(w), G^{B_j}_{d_{B_j}}(w)) \cdot \Delta T(B_i, B_j)}{|\Delta I(G^{B_i}_{d_{B_i}}(w), G^{B_j}_{d_{B_j}}(w))||\Delta T(B_i, B_j)|}, \tag{13}$$

$$\mathcal{L}_{domain-norm}(D_\varphi, B_i) = \|D_\varphi(E_T(t_{B_i})) - 1\|^2 \tag{14}$$

Then the overall training loss for the HDN $D_\varphi(\cdot)$ is

$$\mathcal{L}(\varphi) = \lambda_{direction} \sum_{i=1}^{m} \mathcal{L}_{direction}(G^{B_i}_{D_\varphi(E_T(t_{B_i}))}(w), G^A(w), B_i, A) + \tag{15}$$

$$+ \lambda_{tt-direction} \sum_{i \neq j}^{m} \mathcal{L}_{tt-direction}(G^{B_i}_{D_\varphi(E_T(t_{B_i}))}(w), G^{B_j}_{D_\varphi(E_T(t_{B_j}))}(w), B_i, B_j) +$$

$$+ \lambda_{domain-norm} \sum_{i=1}^{m} \mathcal{L}_{domain-norm}(D_\varphi, B_i) \tag{16}$$

### A.2.2 Architecture of the HDN

We use the standard ResNet-like architecture for the HDN. It has the backbone part which has 10 ResBlocks and the part that consists of 17 heads. The number of heads equals the number of StyleGAN2 layers in the synthesis network $G_{sys}$. Each head has 5 ResBlocks and outputs the domain vector for the corresponding StyleGAN2 layer. We illustrate the overall architecture of the HDN in Figure 6. It has 43M parameters. We use the same architecture for all experiments.

### A.2.3 Inference Time

The inference time of the HDN network on 1 Tesla A100 GPU is almost the same as the one forward pass through StyleGAN2 generator which works in 0.02 seconds.

### A.2.4 Training on Fixed Number of Domains

For training the HDN on fixed number of domains we use the loss function from Equation (16). As training target domains we take the following 20 domains (we provide in the format "the target domain - the corresponding source domain"):

1. Anime Painting - Photo
2. Impressionism Painting - Photo
3. Mona Lisa Painting - Photo
4. 3D Render in the Style of Pixar - Photo
5. Painting in the Style of Edvard Munch - Photo
6. Cubism Painting - Photo
7. Sketch - Photo
8. Dali Painting - Photo
9. Fernando Botero Painting - Photo
10. A painting in Ukiyo-e style - Photo

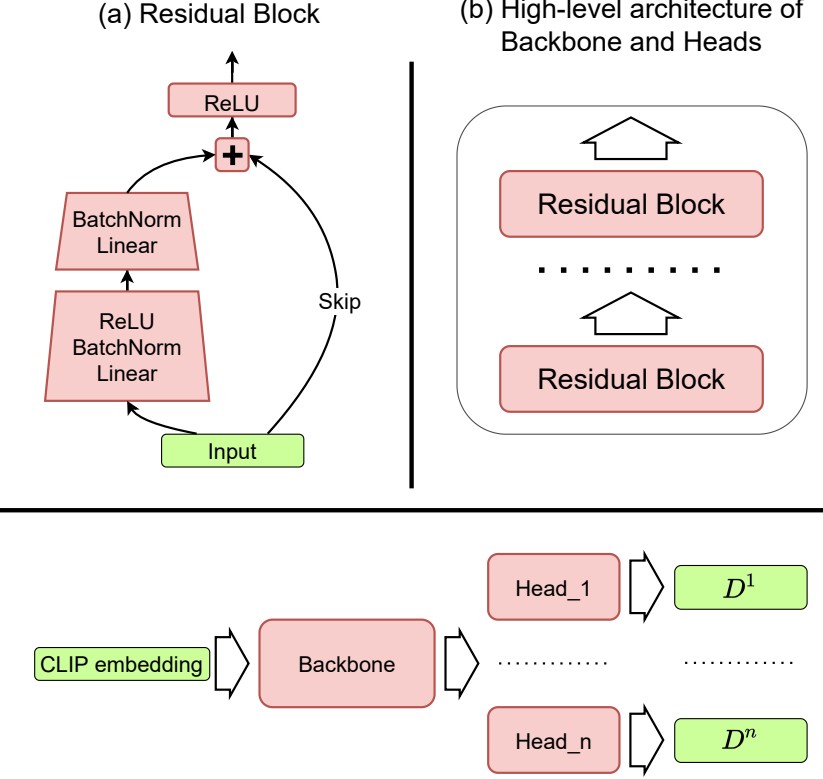

(a) Residual Block

(b) High-level architecture of Backbone and Heads

(c) High-level architecture of HDN

Figure 6: Detailed HDN architecture diagram. (a) - base residual block which is included into backbone and head parts of the HDN. (b) - the detailed backbone and head architecture, each module use the same sequence of ResBlocks. (c) - the detailed architecture of the HDN with data flow.

11. Tolkien Elf - Human

12. Neanderthal - Human

13. The Shrek - Human

14. Zombie - Human

15. The Hulk - Human

16. The Thanos - Human

17. Werewolf - Human

18. Nicolas Cage - Human

19. The Joker - Human

20. Mark Zuckerberg - Human

**Hyperparameters** For 20 descriptions in training list following setup is considered. The HDN trained for 1000 number of iterations. Batch size 24 is used. Weights of the terms from Equation (16) as follows: $\lambda_{direction} = 1.0$, $\lambda_{tt-direction} = 0.4$, $\lambda_{domain-norm} = 0.8$. We use two Vision-Transformer based CLIP models, "ViT-B/32" and "ViT-B/16". To optimize HDN we use an ADAM Optimizer with betas= $(0.9, 0.999)$, learning rate= $5e-5$, weight decay= $0$.

### A.2.5   Training Time

The training time of the HDN on 20 domains for 1000 iterations on single Tesla A100 GPUs takes about 2 hours.

### A.2.6   Ablation Study on the Loss Terms

We perform both the quantitative and qualitative ablation study on the domain-norm and tt-direction loss terms that are defined in Appendix A.2.1.

For the qualitative analysis we consider three domains (Anime Painting, Mona Lisa Painting, A painting in Ukiyo-e style) for the HyperDomainNet that was trained on 20 different domains (see the full list in Appendix A.2.4). We provide the visual comparison for these domains with respect to the using loss terms in the training loss of the HyperDomainNet (see Figure 7). We can see that without additional loss terms the model considerably collapses within each domain. After adding domain-norm it solves the problem of collapsing within domains but it starts mix domains with each other, so we obtain the same style for different text descriptions. And after using tt-direction loss eventually allows us to train the HyperDomainNet efficiently on these domains without collapsing.

For the quantitative results we use the metrics Quality and Diversity that were introduced in Appendix A.3.3. The results are provided in Table 4. We see that the initial model without loss terms obtains good Quality but very low Diversity. The domain-norm significantly improves the diversity in the cost of degrading the Quality. The tt-direction provides a good balance between these two metrics which we also we qualitatively in Figure 7.

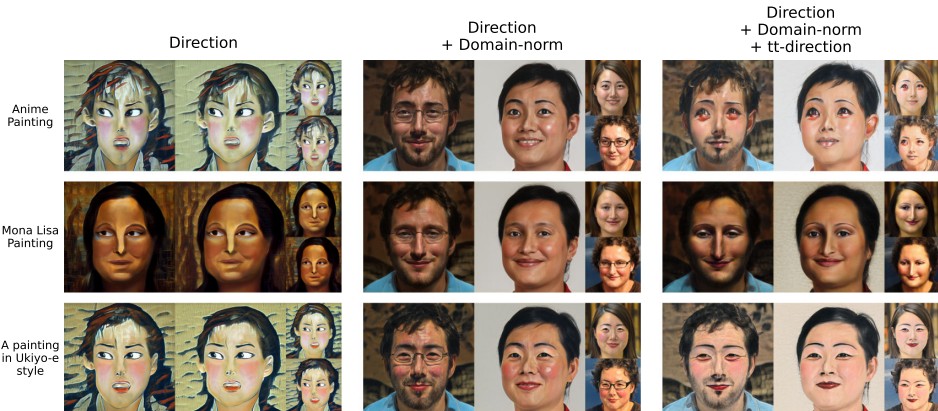

Figure 7: Ablation study on the loss terms of the HyperDomainNet.

**Additional Samples**   We show results for the first 10 domains in Figure 5. The next 10 domains we provide in Figure 8.

Table 4: Ablation study on the loss terms of the HyperDomainNet.

| Model | Quality | Diversity |
|---|---|---|
| Anime Painting | | |
| Multi-Domain | 0.271 | 0.128 |
| Multi-Domain+domain_norm | 0.210 | 0.338 |
| Multi-Domain+domain_norm+tt_direction | 0.260 | 0.256 |
| Zombie | | |
| Multi-Domain | 0.254 | 0.079 |
| Multi-Domain+domain_norm | 0.246 | 0.203 |
| Multi-Domain+domain_norm+tt_direction | 0.258 | 0.191 |
| Across ten domains | | |
| Multi-Domain | $0.275 \pm 0.035$ | $0.099 \pm 0.026$ |
| Multi-Domain+domain_norm | $0.218 \pm 0.026$ | $0.306 \pm 0.040$ |
| Multi-Domain+domain_norm+tt_direction | $0.247 \pm 0.026$ | $0.250 \pm 0.041$ |

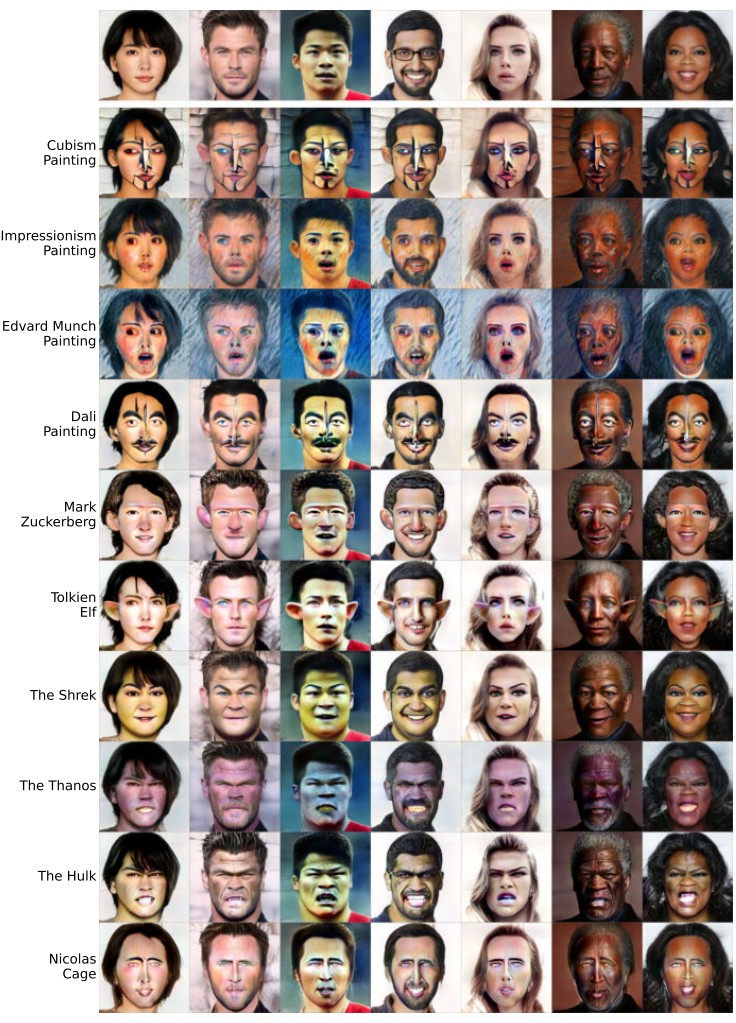

Figure 8: Other domains which were included into train description list from left block of Figure 5.

### A.2.7 Training on Potentially Arbitrary Number of Domains

Improving the generalization ability of the HDN is challenging because it tends to overfit on training domains and for unseen ones it predicts domains that are very close to train ones. One way to tackle this problem is to considerably extend the training set. For this purpose, we use three techniques: (i) generating many training domains by taking combinations of different ones; (ii) sampling CLIP embeddings from convex hull of the initial training embeddings; (iii) resample initial CLIP embeddings given cosine similarity. We discuss each technique further.

**Generating Training Domains by Taking Combinations**    We can describe domains by indicating different properties of the image such as image style (e.g. Impressionism, Pop Art), image type (e.g. Photo, Painting), artist style (e.g. Modigliani). Also we can construct new domains by taking combinations of these properties (e.g. Modigliani Painting, Impressionism Photo). So, we take 32 image styles, 13 image types and 7 artists and by taking all combinations of these properties we come up with 1040 domains. We provide the full list of properties we use in our training:

- image styles: 'Pop Art', 'Impressionism', 'Renaissance', 'Abstract', 'Vintage', 'Antiquity', 'Cubism', 'Disney', 'Chinese', 'Japanese', 'Spanish', 'Italian', 'Dutch', 'German', 'Surreal', 'WaltDisney', 'DreamWorks', 'Modern', 'Realism', 'Starry Night', 'Old-timey', 'Pencil', 'Gouache', 'Acrylic', 'Watercolor', 'Oil', 'Black', 'Blue', 'Charcoal', 'Manga', 'Kodomo';

- image types: 'Portrait', 'Image', 'Photo', 'Painting', 'Graffiti', 'Photograph', 'Cartoon', 'Stereo View', 'Drawing', 'Graphics', 'Mosaic', 'Caricature', 'Animation';

- artists: 'Raphael', 'Salvaror Dali', 'Edvard Munch', 'Modigliani', 'Van Gogh', 'Claude Monet', 'Leonardo Da Vinci'

The algorithm of generating combinations is

```python
def get_train_descriptions(
    base_image_types: list,
    base_image_styles: list,
    artists: list
):
    result_target_domains = []

    for stylization, image_type in product(base_image_styles + artists, base_image_types):
        result_target_domains.append(f"{stylization} {image_type}")

    return result_target_domains
```

Figure 9: The algorithm of generating combinations implemented on **Python**.

**Generating CLIP Embeddings from Convex Hull**    In the usual training of the HDN as in Appendix A.2.4 we use the CLIP embeddings of the target domains $t_{B_1} = E_T(B_1), \ldots, t_{B_m} = E_T(B_m)$. To cover more CLIP space we propose to use new embeddings $t'_{B_1}, \ldots, t'_{B_m}$ from the convex hull of the initial ones:

$$t'_{B_i} = \sum_{j=1}^{m} \alpha_j t_{B_j}, \ i = 1, \ldots, m, \tag{17}$$

where $\alpha_1, \ldots, \alpha_m \sim \text{Dir}(\beta)$ (Dirichlet distribution), $\sum_{i=1}^{m} \alpha_i = 1, \ \alpha_i \geqslant 0, i = 1, \ldots, m.$ (18)

We use $\beta = \dfrac{1}{batch\ size}$.

**Resampling Initial CLIP Embeddings Given Cosine Similarity**    To further extend the CLIP space we cover during training of the HDN we resample initial CLIP embeddings of the target domains $t_{B_1}, \ldots, t_{B_m}$ constrained to the cosine similarity. So, before generating from convex hull we replace the initial embeddings by new ones $\hat{t}_{B_1}, \ldots, \hat{t}_{B_m}$ such that $\cos(t_{B_1}, \hat{t}_{B_1}) = \gamma$. To obtain

these embeddings we use the following operation:

$$\hat{t}_{B_i} = \text{resample}(t_{B_i}), \ i = 1, \ldots, m, \tag{19}$$

$$\text{where resample}(t_{B_i}) = t_{B_i} \cdot \cos\gamma + \text{norm}(v - proj_{t_{B_i}} v) \cdot \sin\gamma, \tag{20}$$

$$v \sim \mathcal{N}(v|0, \mathbf{I}), \quad \text{norm}(u) = \frac{u}{||v||_2} \tag{21}$$

It allows us to cover the part of the CLIP space outside of the initial convex hull. We observe that it improves the generalization ability of the HDN.

**Hyperparameters**    We train the HDN for 10000 number of iterations. We use batch size of 96. We set weights of the terms from Equation (16) as follows: $\lambda_{direction} = 1.0$, $\lambda_{tt-direction} = 0.4$, $\lambda_{domain-norm} = 0.8$. We use two Vision-Transformer based CLIP models, "ViT-B/32" and "ViT-B/16". To optimize HDN we use an ADAM Optimizer with betas= $(0.9, 0.999)$, learning rate= $5e-5$, weight decay= 0.

**Training Time**    The training time of the HDN for 10000 iterations on 4 Tesla A100 GPUs takes about 50 hours.

**Additional Samples**    Additional samples of unseen domains for the HDN is demonstrated in Figure 10.

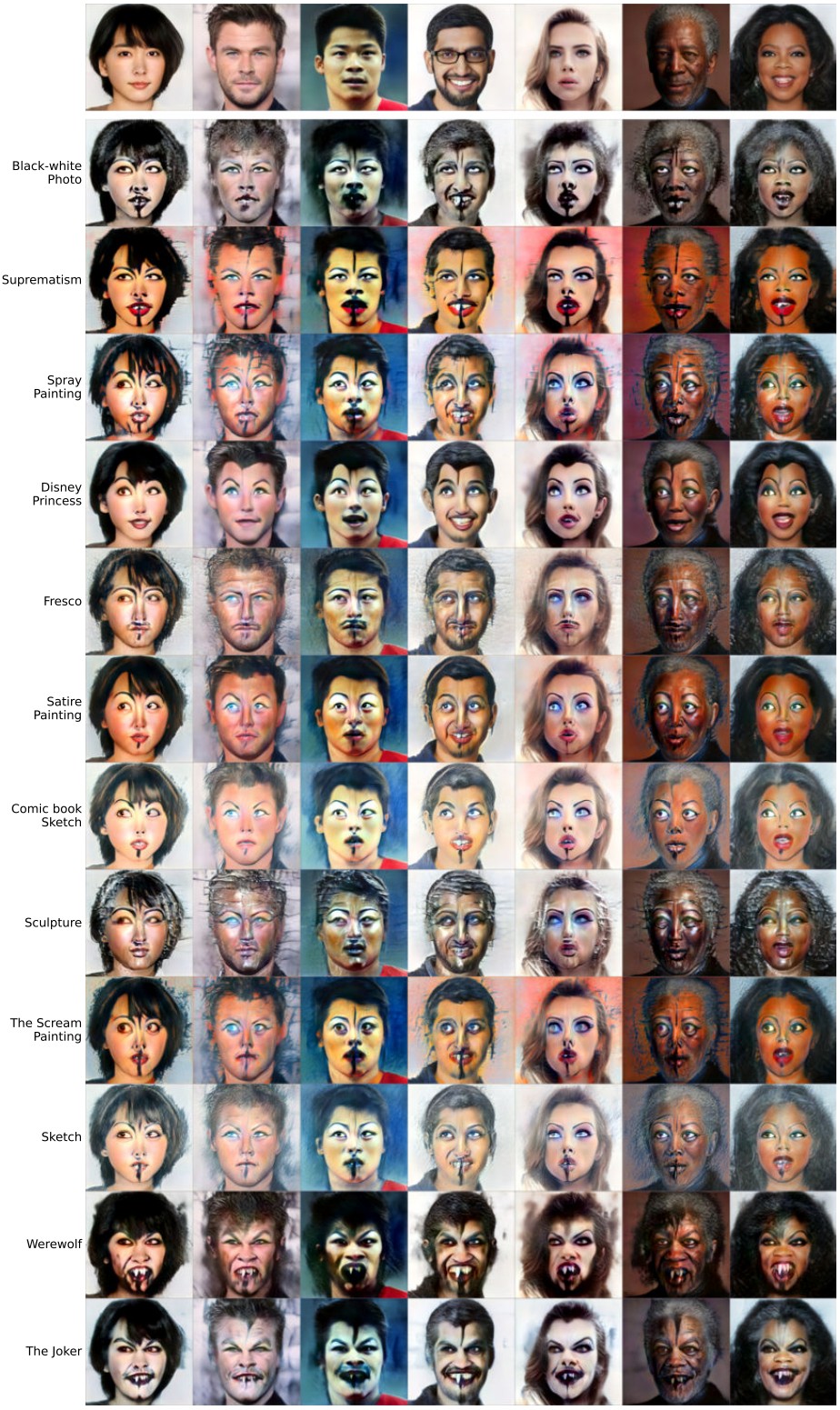

Figure 10: Other visual results for descriptions which were not included into training list during HDN training.

### A.3 Results for Text-Based Domain Adaptation

### A.3.1 Hyperparameters

StyleGAN-NADA with our parameterization trained for 600 iterations with batch size of 4. Style mixing probability is set to 0.9, the weight of the $\mathcal{L}_{direction}$ is 1.0 and $\mathcal{L}_{indomain-angle}$ is 0.5 and ADAM optimizer with betas= $(0., 0.999)$, learning rate= 0.002, weight decay= 0.

For the original StyleGAN-NADA [6] number of iterations is decreased to 200 because for more iterations it starts to collapse.

"ViT-B/32" and "ViT-B/16" CLIP Vision-Transformed models used in all setups.

### A.3.2 Training and Inference Time

The training of the one target domain for 600 iterations on a single Tesla A100 GPU takes about 15 minutes on batch size 4.

The inference time consists of two parts. The first one is the embedding process of the real image which takes 0.23 seconds using ReStyle [1]. The second part is the forward pass through adapted GAN generator which works in 0.02 seconds.

### A.3.3 Quantitative Results

We provide the quantitative comparison for the text-based domain adaptation by evaluating the "Quality" and "Diversity" metrics in a straightforward way.

As the "Quality" metric we estimate how close the adapted images to the text description of the target domain. That is we compute the mean cosine similarity between image CLIP embeddings and the embedding of the text description:

$$\text{Quality} = \frac{1}{n} \sum_{i=1}^{n} \langle E_T(\text{target\_text}), E_I(I_i) \rangle, \text{ where} \tag{22}$$

$n$ - number of the generated adapted images (we use 1000),

$E_T$ - text CLIP encoder,

$E_I$ - image CLIP encoder,

$I_1, \ldots, I_n$ - generated adapted images.

As $E_I$ encoder we use only ViT-L/14 image encoder that is not applied during training (in the training we use ViT-B/16, ViT-B/32 image encoders).

As the "Diversity" metric we estimate the mean pairwise cosine distance between all adapted images:

$$\text{Diversity} = \frac{2}{n(n-1)} \sum_{i<j}^{n} (1 - \langle E_I(I_i), E_I(I_j) \rangle), \text{ where} \tag{23}$$

$n$ - number of the generated adapted images (we use 1000),

$E_I$ - image CLIP encoder,

$I_1, \ldots, I_n$ - generated adapted images.

We compute these two metrics for the ten text domains: Anime Painting, Mona Lisa Painting, 3D Render Pixar, Sketch, Ukiyo-e Painting, Fernando Botero Painting, Werewolf, Zombie, The Joker, Neanderthal. We separately report metrics for two domains Anime Painting and Zombie to better reflect the metrics behaviour. Also we report the overall metrics across all nine domains. The results are provided in Table 5.

From these results we see that our model performs comparably with the StyleGAN-NADA with respect to Quality while having better Diversity. Also we can observe that the indomain angle loss significantly improves the Diversity for both models StyleGAN-NADA and Ours while lightly decreases the Quality.

For the multi-domain adaptation model we see that it has lower diversity than StyleGAN-NADA and Ours and comparable Quality while being adapted to all these domains simultaneously.

Also we report samples for the StyleGAN-NADA and our model with and without indomain angle loss in Figures 11 and 12. We see that qualitatively indomain angle loss also significantly improves the diversity of the domain adaptation methods.

### A.3.4 Additional Samples

We show additional domains for FFHQ dataset in Figure 13. Also we demonstrate how our method works on another datasets such as LSUN Church in Figure 14, LSUN Cats in Figure 15, and LSUN Cars in Figure 16.

Table 5: Evaluation of text-based adaptation methods.

| Model | Quality | Diversity |
|---|---|---|
| **Anime Painting** | | |
| StyleGAN-NADA [6] | 0.289 | 0.244 |
| Ours | 0.284 | 0.305 |
| StyleGAN-NADA+indomain | 0.256 | 0.401 |
| Ours+indomain | 0.251 | 0.404 |
| Multi-Domain+domain_norm+tt_direction | 0.260 | 0.256 |
| **Zombie** | | |
| StyleGAN-NADA [6] | 0.257 | 0.153 |
| Ours | 0.264 | 0.275 |
| StyleGAN-NADA+indomain | 0.261 | 0.354 |
| Ours+indomain | 0.247 | 0.372 |
| Multi-Domain+domain_norm+tt_direction | 0.258 | 0.191 |
| **Across ten domains** | | |
| StyleGAN-NADA [6] | $0.270 \pm 0.032$ | $0.196 \pm 0.034$ |
| Ours | $0.256 \pm 0.019$ | $0.306 \pm 0.030$ |
| StyleGAN-NADA+indomain | $0.249 \pm 0.018$ | $0.394 \pm 0.026$ |
| Ours+indomain | $0.240 \pm 0.018$ | $0.398 \pm 0.026$ |
| Multi-Domain+domain_norm+tt_direction | $0.247 \pm 0.026$ | $0.250 \pm 0.041$ |

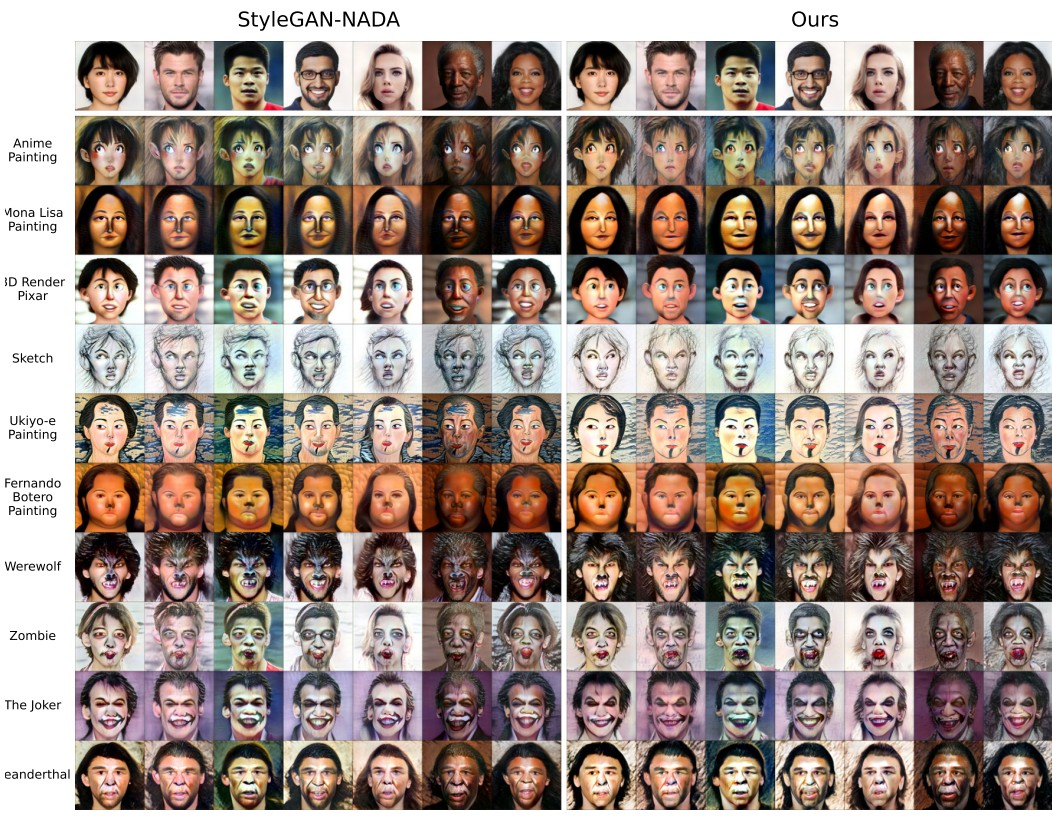

Figure 11: Comparison of text-based domain adaptation methods without indomain angle loss. Left column represents StyleGAN-NADA [6], right column represents our model.

StyleGAN-NADA (w indomain)          Ours (w indomain)

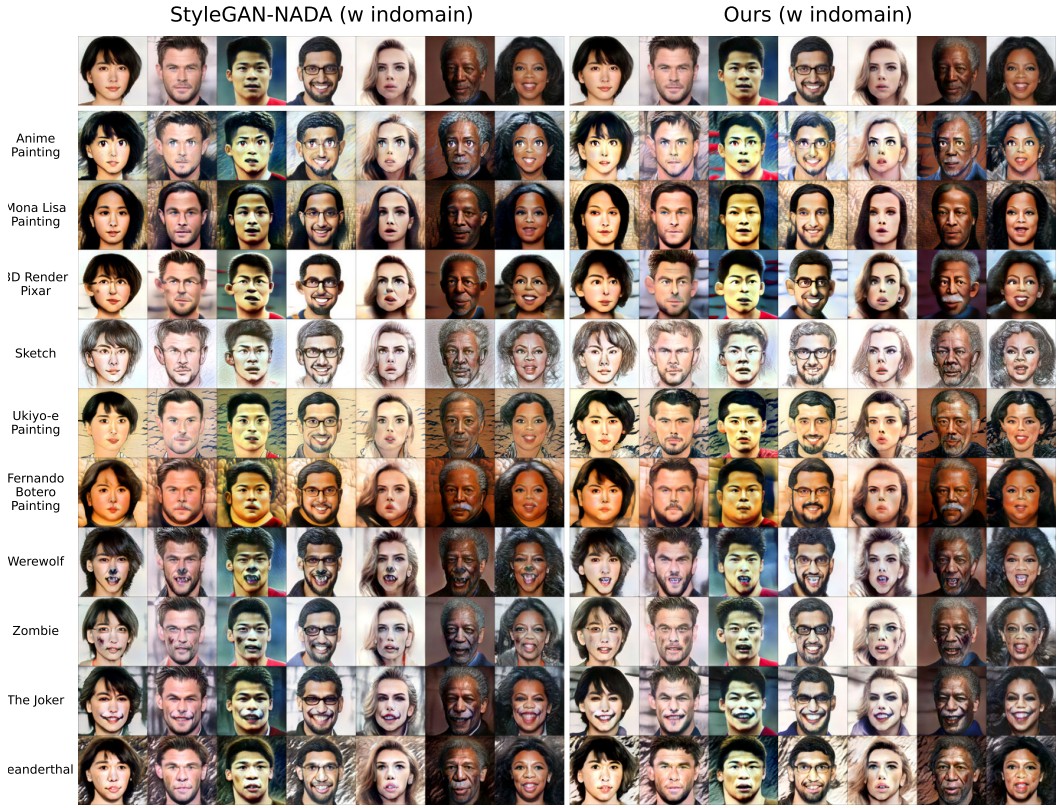

Anime
Painting

Mona Lisa
Painting

3D Render
Pixar

Sketch

Ukiyo-e
Painting

Fernando
Botero
Painting

Werewolf

Zombie

The Joker

Neanderthal

Figure 12: Comparison of text-based domain adaptation methods with indomain angle loss. Left column represents StyleGAN-NADA [6], right column represents our model.

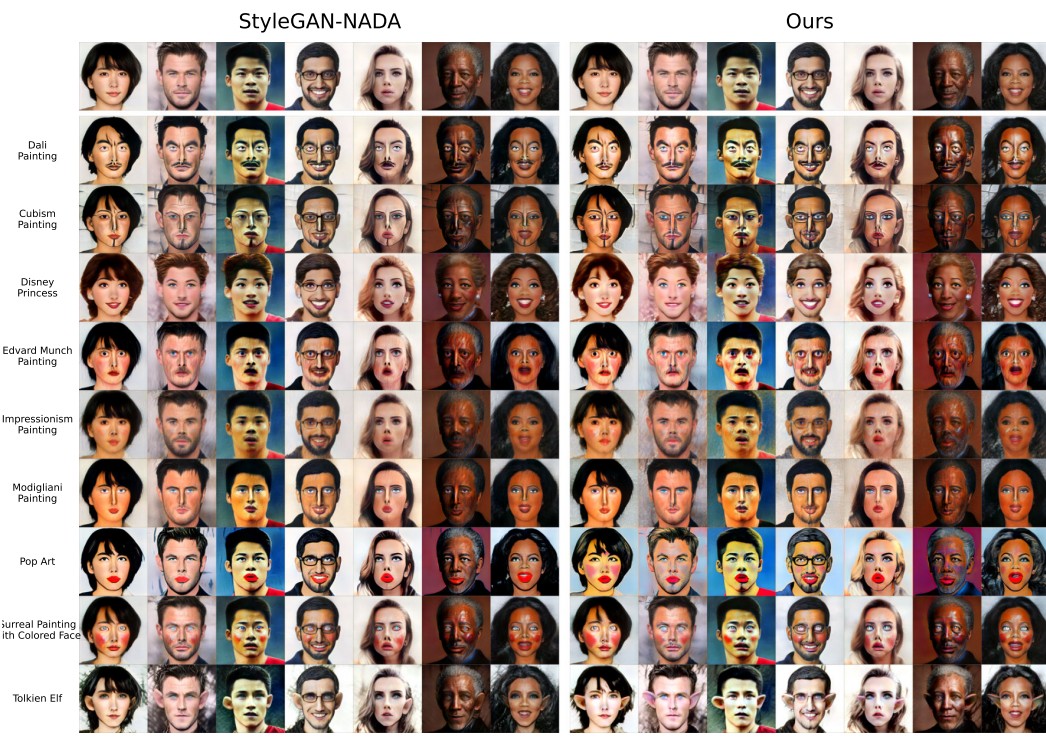

Figure 13: Comparison for other domains in single domain adaptation setup on real images. Left column represents StyleGAN-NADA [6], right column represents results with same approach patched with ours parameterization.

StyleGAN-NADA                                    Ours

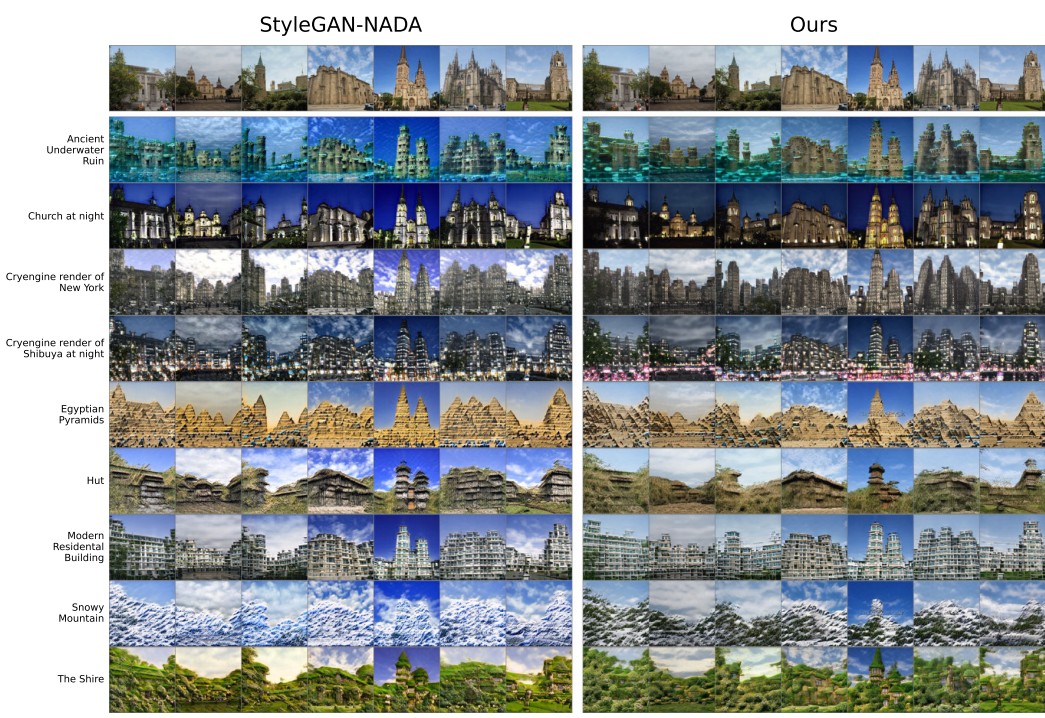

Figure 14: Single domain adaptation comparison for LSUN Church dataset.

StyleGAN-NADA                                    Ours

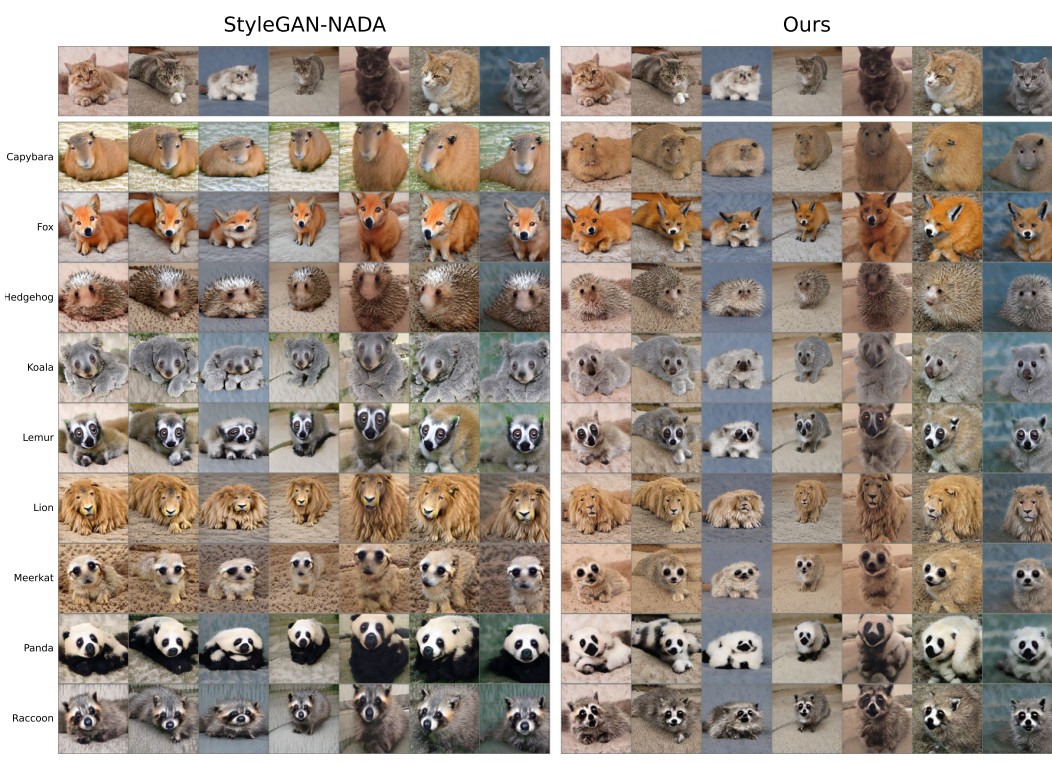

Figure 15: Single domain adaptation comparison for LSUN Cats dataset.

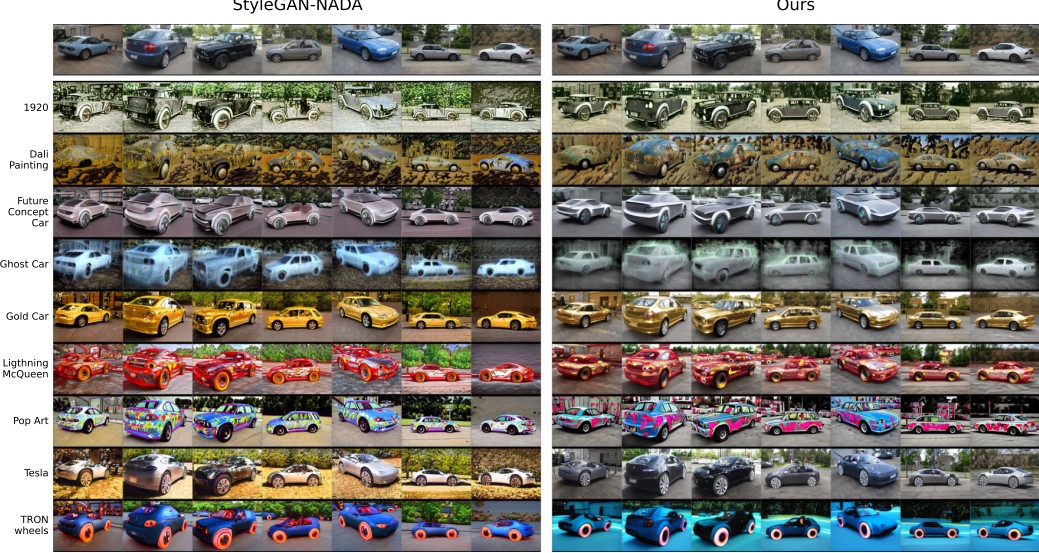

Figure 16: Single domain adaptation comparison for LSUN Cars dataset.

## A.4 Results for One-Shot Domain Adaptation

### A.4.1 Hyperparameters

For each target style image we adapt the generator for 600 iterations as in [48]. We use batch size of 4, fine-tune all layers of the StyleGAN2, set the mixing probability to 0.9. We use all loss terms as in [48] with the same weights and add the $\mathcal{L}_{indomain-angle}$ term with weight 2. For all experiments, we use an ADAM Optimizer with a learning rate of 0.002.

### A.4.2 Training and Inference Time

The training of the one target style image for 600 iterations on a single Tesla A100 GPU takes about 20 minutes. The same as for the text-based adaptation the inference time consists of two parts: embedding process and the forward pass through the generator. The embedding process takes 0.36 seconds for e4e [32] and two minutes for II2S [49]. The second part is the forward pass through adapted GAN generator which works in 0.02 seconds.

### A.4.3 Additional Samples

We provide additional samples in Figures 17 and 18. Also we provide results for other baseline methods in Figure 19.

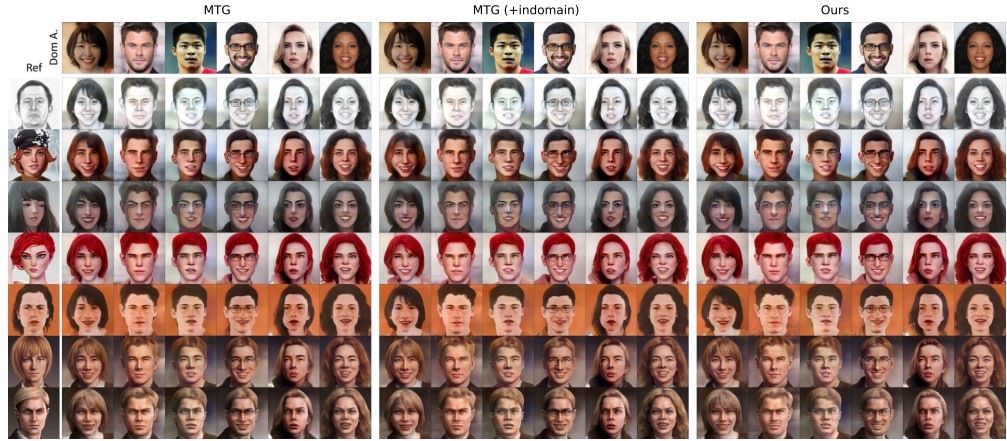

Figure 17: Comparison of one-shot domain adaptation methods: original MindTheGap [48] (left), MindTheGap + indomain (center) and MindTheGap with our parameterization (right).

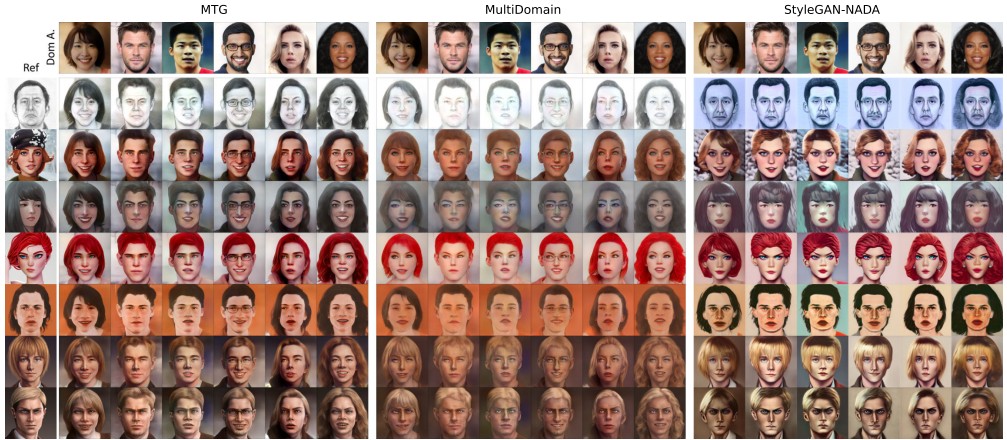

Figure 18: Comparison of one-shot domain adaptation methods: original MindTheGap [48] (left), Multi-Domain model (center) and StyleGAN-NADA (right).

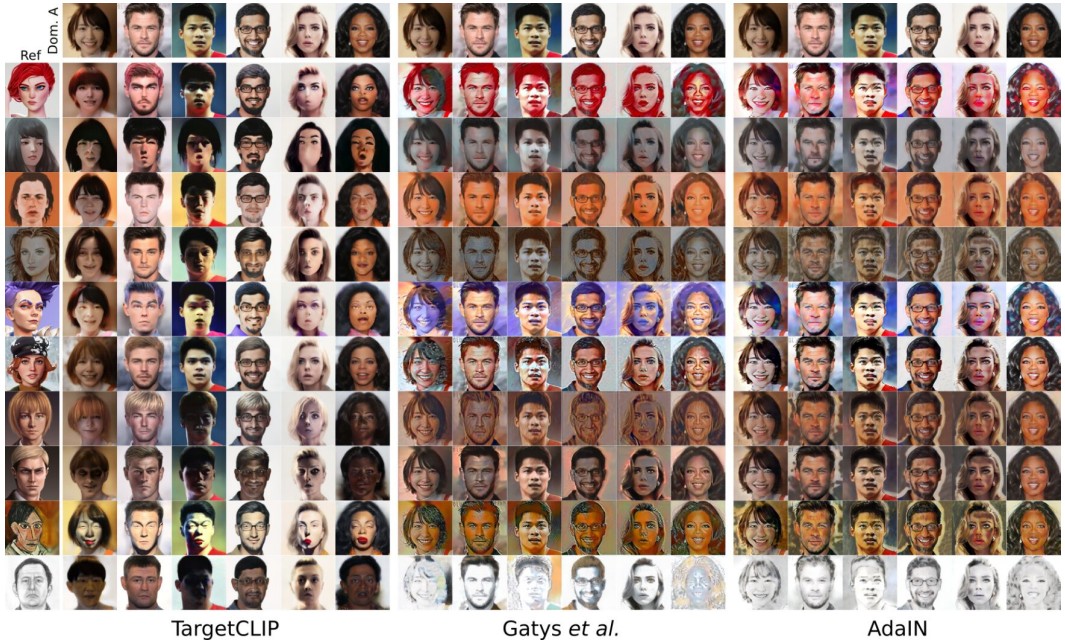

TargetCLIP       Gatys *et al.*       AdaIN

Figure 19: Additional comparisons with other baseline methods including TargetCLIP [4], Gatys et al. [7], and AdaIN [11]. Compare these results to our method in Figure 4. We can see that both the original MindTheGAP and with our parameterization has fewer artifacts.