# OpenReview forum: "HyperDomainNet: Universal Domain Adaptation for Generative Adversarial Networks"
_NeurIPS.cc/2022/Conference — NeurIPS 2022 Accept_

### Official Review · Reviewer_nhQm · 2022-07-10

**Rating:** 6
**Confidence:** 4
**Soundness:** 3 good
**Presentation:** 3 good
**Contribution:** 2 fair

**Summary:**

Inspired by AdaIN, this paper proposes a domain modulation technique to greatly reduce the trainable parameters (from 30 millions to about 6000) for domain adaptation of the StyleGAN generators. In addition to only using $L_{direction}$ (or $L_\text{clip-across}$), this paper also introduces a new in-domain angle consistency loss $L_{indomain-angle}$ as a regularization term to improve the diversity of the fine-tuned generator. To address the multi-domain adaptation problem,  the paper proposes a hyper network to predict the domain parameters given an input target image.

**Questions:**

I think the idea of this paper is good. My main concern is the effectiveness of the proposed method. The paper also lacks fair comparison results. If the author can clarify the questions I asked, I may improve the score.


**Limitations:**

Adequate

**Strengths And Weaknesses:**

The paper is well written and easy to follow. I like the idea of reducing the trainable parameters in fine-tuning and further using a hypernet to predict domain parameters. Compared with previous optimization-based fine-tuning methods, this proposed approach is more suitable for real-time applications.

However, my concern is whether the proposed method is actually effective or can achieve similar results to prior works. The visual results presented in the paper are not good enough. As far as I know, the results of MTG should also be better than what is shown in the paper. Why skip the MTG results in the original paper and mark the "MTG+indomain" results as MTG results? Why not add the HyperDomainNet results to the visual comparison in Fig 4?

Another question is about the regularization term proposed in the paper. $L_{indomain-angle}$ is very similar to the $L_\text{clip-within}$ proposed in MTG, except that the cos similarity loss is replaced by the l2 loss. There is no evidence that using l2 loss produces better results.

---

> ### Author Response · Authors · 2022-08-02
> **Response to nhQm**
>
> Dear Reviewer nhQm,
>
> Thank you for your thoughtful feedback and constructive comments. We have updated the paper to address your concerns and comment on them below.
>
> Regarding the observation that “MTG should also be better than what is shown in the paper”, we agree with you. However, to obtain these results we used the official code of the MTG method and it was difficult to achieve the same results as in the MTG paper. As we understand the visual quality is very sensitive to the hyperparameters of the image inversion method II2S that is used for the inversion of celebrity faces and the style images. We use the default hyperparameters that give us results that were visually less satisfactory than the original MTG results.
>
> But after more careful tuning of II2S’s hyperparameters (mainly the learning rate) we achieved much better visual quality on par with the results reported in the original MTG. So, we updated our results in Figure 4. It can be seen that our method can achieve visual quality comparable with the original MTG, so it proves its efficiency for the one-shot domain adaptation.
>
> Also we want to clarify why we showed results only for the "MTG+indomain" method and skipped the results of the original MTG. The main reason is that when we obtain results for the MTG and MTG+indomain we observe that they have comparable visual quality but the MTG+indomain has better quantitative metrics. So, we considered the MTG+indomain as the more challenging baseline and skipped the original MTG results. But we understand your concern and add MTG results (reproduced by ourselves) in Figure 15 (Appendix A.4.3) for more transparent comparison.
>
> > Why not add the HyperDomainNet results to the visual comparison in Fig 4?
>
> For the HyperDomainNet we mainly focus on the text-based domain adaptation to show its effectiveness and ability to be trained on a number of diverse target domains simultaneously. The application of the HyperDomainNet to the image-based one-shot domain adaptation is also an interesting and challenging problem. We did not consider it in the scope of this paper. However, to demonstrate that it is also a perspective and feasible direction of research we obtained the preliminary results for such setting (see in Figure 16, Appendix A.4.3).
>
> Regarding the similarity of $L_{indomain−angle}$ and $L_{clip-within}$ losses, we agree that it is true to some extent, however there are several crucial differences.
>
> Before we describe the differences let us provide the definitions of these losses:
> $$
> L_{indomain-angle} = \sum\limits_{i<j}^n (\langle I^A_i, I^A_j \rangle - \langle I^B_i, I^B_j \rangle)^2, \\\\
> L_{clip-within} = \sum\limits_i^n \left(1 - \dfrac{\langle \Delta I^A_i, \Delta I^B_i \rangle}{\|\Delta I^A_i\|\|\Delta I^B_i\|}\right), \text{ where } \\\\ n \text{ - batch size, } \\\\
> I^A_1, \dots, I^A_n \text{ - CLIP embeddings of the source domain images, } \\\\
> I^B_1, \dots, I^B_n \text{ - CLIP embeddings of the target domain images, } \\\\
> \Delta I^A_i = I^A_i - I^A_*, \; I^A_* \text{ - CLIP embedding of the reference image in the source domain, } \\\\
> \Delta I^B_i = I^B_i - I^B_*, \; I^B_* \text{ - CLIP embedding of the reference image in the target domain, }
> $$
>
> The main differences:
> - $L_{indomain-angle}$ preserves **pairwise** cosine distances between **all images** from the batch before and after domain adaptation. While $L_{clip-within}$ cares only the directions between the reference image and other images from the batch. So, $L_{clip-within}$ does not consider pairwise alignment of the images after domain adaptation which can be less effective against collapsing.
> - In $L_{indomain-angle}$ we consider pairwise cosine distances of the image CLIP embeddings. In contrast, $L_{clip-within}$ utilizes cosine distances of the vectors $\Delta I^A_i, \Delta I^B_i$ that are differences between CLIP embeddings. We note that the CLIP cosine distance is a natural distance only for the vectors that represents CLIP embeddings of the real image or text. Therefore, we avoid computing the cosine distance between the vectors $\Delta I^A_i, \Delta I^B_i$ and use $L_2$ loss to force the same pairwise cosine distances between CLIP embeddings of the images.
> - To compute $L_{clip-within}$ it is required to have embeddings $I^A_*, I^B_*$ of the reference image in the source and target domains unlike $L_{indomain-angle}$. This limitation is more crucial in the case of text-based domain adaptation where it is not obvious how to find the concise text description of the source domain that will be correspond to the description of the target domain. At the same time,  $L_{indomain-angle}$ can be easily applied to the text-based setting.
>
> So, considering the discussed distinctions we can say that $L_{indomain-angle}$ is sufficiently different from $L_{clip-within}$ loss.
>
> Also in experiments we quantitatively show that adding $L_{indomain-angle}$ significantly improves FID and Precision metrics (see Table 1.).

---

### Official Review · Reviewer_M94K · 2022-07-11

**Rating:** 5
**Confidence:** 3
**Soundness:** 2 fair
**Presentation:** 2 fair
**Contribution:** 2 fair

**Summary:**

The few-shot domain adaptation of GAN generally achieve adaptation to the target domain by fine tuning the generator that trained on source domain with only a few data from target domain. However fine-tuning GAN could be extremely time consuming and resource demanding. Considering the application scenario that the discrepancy among domains is low, fine-tuning the whole network might be redundant and unnecessary. In this work, the authors proposed to incorporate a novel domain-modulation operator to each convolution layers such that the model can be adapted to target domain by optimizing a small domain-specific vector. Further, they developed a hyper-network to predict the domain-specific vector to support adapting to multiple domains simultaneously. They conducted extensive experiments to empirically demonstrate the effectiveness of the proposed method.

**Questions:**

- In the supplementary, eq(13), in order to compute the target-target directional loss, are we going to consider all combinations and sum them up? If this is the case, I think the computational cost will exponentially increase with the number of domains increase. In line 271, "training on potentially arbitrary number of domains" is it realistic to handle a large number of domains, i.e millions of domains? In line 273-274, "fix large number of domains (several hundreds)" I can imagine these would be tens of thousands of target-target combinations, what is the computational cost of this case?
- I assume the motivation of tt-direction loss and domain-norm loss is to avoid the model collapse. However I failed to see why these losses are able to prevent the representations of domains overlap with each other. I think tt-direction loss essentially encourage the representations of text from different domains are aligned with the representations of image from different domains. I don't think it prompt the model push apart domains. I would very much appreciate if the author share their insights on this to help me to understand.

**Limitations:**

A more detailed description of their method and more insightful explanation would make the paper more readable.

**Strengths And Weaknesses:**

Strengths:
- Their framework reduced the number of trainable parameters significantly compared with fine-tuning techniques, as the model weights are adapted to target domain by projecting with a domain-specific vector.
- The proposed HyperDomainNet enable the model to adapt to multiple domains simultaneously by predicting the domain-specific adaptation vector.

Weaknesses:
- The abuse of the math notations and the lack of clarification of the shape of each notation. For instance, in eq(7), it is unclear if $\omega_{ijk}$ is a metric or vector. If my understanding is correct, since $s$ and $d$ are vectors, $\omega$ is supposed to be vectors as well.
- The math description could be more concise and precise such that the main body would have sufficient space to include the training objective, instead of put it as supplementary.
- It is better to include more technique details either in the main body or in supplementary to make it self-contained.
- The writing could be further polished. There are a lot of grammatical errors, e.g line 130 $n$ is a batch size, line 248, "allows to adapt the generator to ..."

---

> ### Author Response · Authors · 2022-08-02
> **Response to M94K**
>
> Dear Reviewer M94K,
>
> Thank you for your thoughtful feedback and constructive comments. We will address your concerns regarding a more clear and detailed description of our method in further revisions of our paper. Below we address your questions about target-target directional loss.
>
> > In the supplementary, eq(13), in order to compute the target-target directional loss, are we going to consider all combinations and sum them up? If this is the case, I think the computational cost will exponentially increase with the number of domains increase. In line 271, "training on potentially arbitrary number of domains" is it realistic to handle a large number of domains, i.e millions of domains? In line 273-274, "fix large number of domains (several hundreds)" I can imagine these would be tens of thousands of target-target combinations, what is the computational cost of this case?
>
> Thank you for this question. We think that it is important to clarify this moment. As in Eq. 15 where we provide the overall training loss for the HDN the target-target directional loss part is
>
> $$
> \sum\limits_{i > j}^m \mathcal{L}_{tt-direction}(I^{B_i}, I^{B_j}, B_i, B_j), \text{ where } m \text{ - number of domains}.
> $$
>
> From this formula we see that the number of terms in this sum is $(m^2 - m) / 2$ because we consider all pairwise combinations of domains except the pairs when the domains are the same. Also we should note that we consider only the domains that are taken to the current batch. So, if the size of the batch $n$ is less than $m$ then we will consider $(n^2 - n) / 2$ pairs in our target-target directional loss.
>
> We see that the computational cost will increase quadratically with the batch size that is practically feasible for a sufficiently large number of domains. We should note that in the case of millions of domains the computational cost of the target-target directional loss is negligible concerning other parts of the model (e.g. forward passes of the networks).
>
> To be more concrete let us consider the computational cost of the target-target directional loss for the case of hundred domains. In this case, we use the batch size of 96. So, we should compute the matrix of pairwise distances between domains from this batch and the computational cost is (96^2 - 96) / 2 = 4560 direction loss computations. Each direction loss computation costs several thousand MACs because it depends linearly on the CLIP embedding size (which is 512). So, the overall complexity is approximately 5 million MACs which is negligible compared to the one forward pass of the StyleGAN which takes 143 billion MACs.
>
> > I assume the motivation of tt-direction loss and domain-norm loss is to avoid the model collapse. However I failed to see why these losses are able to prevent the representations of domains overlap with each other. I think tt-direction loss essentially encourage the representations of text from different domains are aligned with the representations of image from different domains. I don't think it prompt the model push apart domains. I would very much appreciate if the author share their insights on this to help me to understand.
>
> Thank you for this question. At first, we want to note that there are two types of model collapse. The first one is when the images are collapsing within each domain. The second type is when different domains overlap with each other. Each of the proposed tt-direction and domain-norm losses is devoted to preventing its own type of the model collapse.
>
> The domain-norm is a replacement for the indomain angle consistency loss which is inefficient in the case of multiple domain adaptation (see a more detailed explanation in 230-232 lines in the paper). It helps to avoid the first type of model collapse because it regularizes how much the generator weights can change. We observe in the practice that significant changes in the generator weights can lead to the collapse of the images within one domain.
>
> The tt-direction loss is devoted to avoiding the second type of model collapse. The idea is the following. As you said correctly “tt-direction loss essentially encourages the representations of text from different domains are aligned with the representations of the image from different domains”. Why does it help the model push apart domains? Let us assume that different domains start overlap each other. It means that representations of images from these domains become much closer than representations of domain texts (because they are not modified). It leads to the misalignment of these image and text representations. So, the tt-direction loss should penalize the model for such behavior.

---

### Official Review · Reviewer_VBeZ · 2022-07-11

**Rating:** 6
**Confidence:** 3
**Soundness:** 2 fair
**Presentation:** 3 good
**Contribution:** 2 fair

**Summary:**

This paper proposes a novel efficient approach for domain adaptation by introducing a domain-modulation operation enabling finetuning by optimizing a compact parameter space. This domain modulating operation is inspired by modulation introduced in StyleGAN2. However, in contrast to StyleGAN2, the only trainable parameters in this paper are domain parameters. They also propose regularization loss based on CLIP embeddings in order to improve the diversity of the generator. They also propose a HyperDomainNet which enables multi-domain adaptation of StyleGAN2. Quantitative results on one-shot adaptation show comparable performance even after a significant reduction in the number of trainable parameters. Qualitative results on Text-Based Domain Adaptation and Multi-Domain Adaptation are comparable to those when entire models were fine-tuned.

**Questions:**

##### Questions
1. Can the authors report quantitative results for Text-Based Domain Adaptation and Multi-Domain Adaptation as well?
2. For HyperDomainNet, there are multiple components in the final training loss (as shown in Appendix A.2.1). Can you perform ablation studies showing the significance of each component in the final training loss?
3. What is the effect of increasing the dimension of domain parameters d? Does having a larger size of d help in the case of datasets with higher source and target domain gaps? Have the authors conducted any experiments on it?

##### Suggestions
1. From lines 197-206: The explanation for collapsing after several hundred iterations would be more rigorous if some reference work on this is provided.
2. In Table 1, the authors have shown the improvement in results by adding in-domain angle consistency loss, but it is only for one-shot adaptation. But the experiments for quantitative results that show the effectiveness of domain-norm regularization in multi-domain adaptation setting is missing.
3. The authors can report results by increasing the dimension of domain parameters d. This would be particularly interesting for datasets with higher source and target domain gaps.

##### Minor comments and typos
1. Line 83: Typcally --> Typically
2. Line 94: consisitency --> consistency
3. Line 153 & Equation 6: clip_accross --> clip_across (same error in multiple places)
4. Line 175: controled --> controlled
5. Line 211: L --> \mathcal{L}
6. Line 231: during --> the
7. Line 256: fisrt --> first

The overall readability of the paper is good. But some of the sentences can be rewritten for better readability.

**Limitations:**

The limitations and potential negative societal impact have been discussed in the paper.

**Strengths And Weaknesses:**

##### Strengths
1. Efficient approach for domain adaptation by utilizing a domain-modulation operation, thereby, reducing the learnable parameter space for each domain by 4-5 thousand times.
2. The proposed parameterization led to very minor degradation for StyleGAN-NADA and MindTheGap.
3. The paper addresses the three problems of text-based domain adaptation, one-shot domain adaptation, and multi-domain adaptation.
4. The idea of domain-modulation is inspired by modulation in Style-GAN2, but in this paper, the domain-modulating vector is the only learnable parameter.

##### Weaknesses
1. Lack of quantitative results for Text-Based Domain Adaptation and Multi-Domain Adaptation.
2. Ablation study for multiple loss terms in HyperDomainNet is missing.
3. Since the number of trainable parameters has drastically decreased, there is a limit to the possible domain gap between the source and the target domain (as also pointed out by the authors).

---

> ### Author Response · Authors · 2022-08-02
> **Response to VBeZ**
>
> Dear Reviewer VBeZ,
>
> Thank you for your thoughtful feedback and constructive comments. We have updated the paper to address your questions. Particularly, we added quantitative results for text-based domain adaptation and multi-domain adaptation as well. Also, we conducted the ablation study for each loss term from the HyperDomainNet loss. Let us answer your questions in more detail below.
>
> > Can the authors report quantitative results for Text-Based Domain Adaptation and Multi-Domain Adaptation as well?
>
> Thank you for this suggestion, it is really important to quantitatively compare methods in both settings. However, for the text-based domain adaptation, there are no standard metrics and a common practice is to compare methods qualitatively [1]. Therefore, initially, we provided quantitative results only for the image-based domain adaptation.
>
> Because there are no existing metrics for this task, we decided to use the most straightforward way to evaluate the “quality” and “diversity” of the text-based domain adaptation method. As the “quality” metric we estimate how close the adapted images are to the text description of the target domain. That is we compute the mean cosine similarity between image CLIP embeddings and the embedding of the text description:
>
> $$
> \text{Quality} = \dfrac{1}{n} \sum\limits_{i=1}^n \langle E_T(\text{target-text}), E_I(I_i) \rangle, \text{ where } \\\\
> n \text{ - number of the generated adapted images (we use 1000)}, \\\\
> E_T \text{ - text CLIP encoder}, \\\\
> E_I \text{ - image CLIP encoder}, \\\\
> I_1, \dots, I_n \text{ - generated adapted images}.
> $$
>
> As $E_I$ encoder we use only ViT-L/14 image encoder that is not applied during training (in the training we use ViT-B/16, ViT-B/32 image encoders).
>
> As the “diversity” metric we estimate the mean pairwise cosine distance between all adapted images:
>
> $$
> \text{Diversity} = \dfrac{2}{n(n-1)} \sum\limits_{i < j}^n (1 - \langle E_I(I_i), E_I(I_j) \rangle), \text{ where } \\\\
> n \text{ - number of the generated adapted images (we use 1000)}, \\\\
> E_I \text{ - image CLIP encoder}, \\\\
> I_1, \dots, I_n \text{ - generated adapted images}.
> $$
>
> We use these two metrics to compare our model with the StyleGAN-NADA in Appendix A.3.3. For the multi-domain adaptation we provide quantitative results in Appendix A.2.6.
>
> The provided quantitative results show that our method is comparable with the StyleGAN-NADA while having much less trainable parameters and also we illustrate the effectiveness of the indomain angle loss for improving diversity. For the multi-domain adaptation, we quantitatively show the importance of each loss term.
>
> > For HyperDomainNet, there are multiple components in the final training loss (as shown in Appendix A.2.1). Can you perform ablation studies showing the significance of each component in the final training loss?
>
> Thank you for this suggestion. We perform both quantitative and qualitative ablation study in Appendix A.2.6. The results clearly show the significance of each loss term.
>
> > What is the effect of increasing the dimension of domain parameters d? Does having a larger size of d help in the case of datasets with higher source and target domain gaps? Have the authors conducted any experiments on it?
>
> Thank you for this question. It is an interesting analysis and we will add such experiments in the next revision of our paper.
>
> Regarding your minor suggestions, comments and typos, we certainly use them to improve our paper in the next revision.
>
> [1] Rinon Gal, Or Patashnik, Haggai Maron, Gal Chechik, and Daniel Cohen-Or. Stylegan-nada: Clip-guided domain adaptation of image generators. arXiv preprint arXiv:2108.00946, 2021.

---

### Meta-Review · Area_Chair_ancn · 2022-08-30

**Recommendation:** Accept
**Confidence:** Certain

**Metareview:**

This paper presents a domain adaptation technique to finetune a GAN's generator by using a small number of domain modulation parameters. This makes finetuning a pre-trained GAN to different domains very efficient, while not sacrificing upon the generation quality.

The paper received generally positive reviews. There were some concerns about missing experimental results and the authors provided those during the rebuttal phase. There were also some concerns about missing quantitative results for the text-based domain adaptation and multi-domain adaptation. This aspects was also satisfactorily discussed in the rebuttal.

Based on the reviews, and the author response and discussion, and my own reading of the paper, I vote for acceptance. However, the authors are advised to take into account the reviewers' feedback and suggestions to improve the camera-ready version.

**Award:**

No

---

### Decision · Program_Chairs · 2022-09-14

Accept